# Exact Gradients for Stochastic Spiking Neural Networks Driven by Rough Signals

**Christian Holberg**
Department of Mathematics
University of Copenhagen
c.holberg@math.ku.dk

**Cristopher Salvi**
Department of Mathematics
Imperial College London
c.salvi@imperial.ac.uk

## Abstract

We introduce a mathematically rigorous framework based on rough path theory to model stochastic spiking neural networks (SSNNs) as stochastic differential equations with event discontinuities (Event SDEs) and driven by càdlàg rough paths. Our formalism is general enough to allow for potential jumps to be present both in the solution trajectories as well as in the driving noise. We then identify a set of sufficient conditions ensuring the existence of pathwise gradients of solution trajectories and event times with respect to the network's parameters and show how these gradients satisfy a recursive relation. Furthermore, we introduce a general-purpose loss function defined by means of a new class of signature kernels indexed on càdlàg rough paths and use it to train SSNNs as generative models. We provide an end-to-end autodifferentiable solver for Event SDEs and make its implementation available as part of the `diffrax` library. Our framework is, to our knowledge, the first enabling gradient-based training of SSNNs with noise affecting both the spike timing and the network's dynamics.

## 1 Introduction

Stochastic differential equations exhibiting event discontinuities (Event SDEs) and driven by noise processes with jumps are an important modelling tool in many areas of science. One of the most notable examples of such systems is that of stochastic spiking neural networks (SSNNs). Several models for neuronal dynamics have been proposed in the computational neuroscience literature with the *stochastic leaky integrate-and-fire* (SLIF) model being among the most popular choices [19, 56]. In its simplest form, given some continuous input current $i_t$ on $[0, T]$, the dynamics of a single SLIF neuron consist of an Ornstein-Uhlenbeck process describing the membrane potential as well as a threshold for spike triggering and a resetting mechanism [33]. In particular, between spikes, the dynamics of the membrane potential $v_t$ is given by the following SDE

$$dv_t = \mu \left( i_t - v_t \right) dt + \sigma dB_t, \tag{1}$$

where $\mu > 0$ is a parameter and $B_t$ is a standard Brownian motion. The neuron spikes whenever the membrane potential $v$ hits the threshold $\psi > 0$ upon which $v$ is reset to 0. Alternatively, one can model the spike times as a Poisson process with intensity $\lambda : \mathbb{R} \to \mathbb{R}_+$ depending on the membrane potential $v_t$. A common choice is $\lambda(v) = \exp((v - \psi)/\beta)$ [50, 26, 27, 24].

A notorious issue for calibrating Event SDEs such as SSNNs is that the implicitly defined event discontinuities, e.g., the spikes, make it difficult to define derivatives of the solution trajectories and of the event times with respect to the network's parameters using classical calculus rules. This issue is exacerbated when the dynamics are stochastic in which case the usual argument relying on the implicit function theorem, used for instance in [6, 25], is no longer valid.

38th Conference on Neural Information Processing Systems (NeurIPS 2024).

## 1.1 Contributions

In this paper, we introduce a mathematically rigorous framework to model SSNNs as SDEs with event discontinuities and driven by càdlàg rough paths, without any prior knowledge of the timing of events. The mathematical formalism we adopt is that of *rough path theory* [38], a modern branch of stochastic analysis providing a robust solution theory for stochastic dynamical systems driven by noisy, possibly discontinuous, *rough signals*. Although Brownian motion is a prototypical example, these signals can be far more irregular (or rougher) than semimartingales [17, 16, 39].

Equipped with this formalism, we proceed to identify sufficient conditions under which the solution trajectories and the event times are differentiable with respect to the network's parameters and obtain a recursive relation for the exact pathwise gradients in Theorem 3.2. This is a strict generalization of the results presented in [6] and [25] which only deal with ordinary differential equations (ODEs). Furthermore, we define *Marcus signature kernels* as extensions of continuous signature kernels [52] to càdlàg rough paths and show their characteristicness. We then make use of this class of kernels indexed on discontinuous trajectories to define a general-purpose loss function enabling the training of SSNNs as generative models. We provide an end-to-end autodifferentiable solver for Event SDEs (Algorithm 1) and make its implementation available as part of the `diffrax` library [28].

Our framework is, to our knowledge, the first allowing for gradient-based training of a large class of SSNNs where a noise process can be present in both the spike timing and the network's dynamics. In addition, we believe this work is the first enabling the computation of exact gradients for classical SNNs whose solutions are approximated via a numerical solver (not necessarily based on a Euler scheme). In fact, previous solutions are based either on surrogate gradients [46] or follow an optimise-then-discretise approach deriving adjoint equations [56], the latter yielding exact gradients only in the scenario where solutions are available in closed form and not approximated via a numerical solver.[1] Finally, we discuss how our results lead to bioplausible learning algorithms akin to *e-prop* [2].

## 2 Related work

**Neural stochastic differential equations (NSDEs)**   The intersection between differential equations and deep learning has become a topic of great interest in recent years. A neural ordinary differential equation (NODE) is an ODE of the form $dy_t = f_\theta(y_t)dt$ started at $y_0 \in \mathbb{R}^e$ using a parametric Lipschitz vector field $f_\theta : \mathbb{R}^e \to \mathbb{R}^e$, usually given by a neural network [5]. Similarly, a neural stochastic differential equation (NSDE) is an SDE of the form $dy_t = \mu_\theta(y_t)dt + \sigma_\theta(y_t)dB_t$ driven by a $d$-dimensional Brownian motion $B$, started at $y_0 \in \mathbb{R}^e$, and with parametric vector field $\mu_\theta : \mathbb{R}^e \to \mathbb{R}^e$ and $\sigma_\theta : \mathbb{R}^e \to \mathbb{R}^{e \times d}$ that are $\text{Lip}^1$ and $\text{Lip}^{2+\epsilon}$ continuous respectively[2]. Rough path theory offers a way of treating ODEs, SDEs, and more generally differential equations driven by signals or arbitrary (ir)regularity, under the unified framework of *rough differential equations* (RDEs) [44, 22]. For an account on applications of rough path theory to machine learning see [4, 14, 51].

**Training techniques for NSDEs**   Training a NSDE amounts to minimising over model parameters an appropriate notion of statistical divergence between a distribution of continuous trajectories generated by the NSDE and an empirical distribution of observed sample paths. Several approaches have been proposed in the literature, differing mostly in the choice of discriminating divergence. SDE-GANs, introduced in [29], use the 1-Wasserstein distance to train a NSDE as a Wasserstein-GAN [1]. Latent SDEs [36] train a NSDE with respect to the KL divergence via variational inference and can be interpreted as variational autoencoders. In [23] the authors propose to train NSDEs non-adversarially using a class of maximum mean discrepancies (MMD) endowed with signature kernels [30, 52]. Signature kernels are a class of characteristic kernels indexed on continuous paths that have received increased attention in recent years thanks to their efficiency for handling path-dependent problems [35, 54, 10, 53, 9, 48, 41]. For a treatment of this topic we refer the interested reader to [4, Chapter 2]. These kernels are not applicable to sample trajectories of SSNNs because of the lack of continuity.

**Backpropagation through NSDEs**   Once a choice of discriminator has been made, training NSDEs amounts to perform backpropagation through the SDE solver. There are several ways to do this. The

---

[1]The point here is actually a little more subtle. It is in fact possible to obtain exact gradients using the adjoint method as long as one uses reversible (or adjoint) solvers in the forward pass.

[2]These are standard regularity conditions to ensure existence and uniqueness of a strong solution.

first option is simply to backpropagate through the solver's internal operations. This method is known as *discretise-then-optimise*; it is generally speaking fast to evaluate and produces accurate gradients, but it is memory-inefficient, as every internal operation of the solver must be recorded. A second approach, known as *optimise-then-discretise*, computes gradients by deriving a backwards-in-time differential equation, the *adjoint equation*, which is then solved numerically by another call to the solver. Not storing intermediate quantities during the forward pass enables model training at a memory cost that is constant in depth. Nonetheless, this approach produces less accurate gradients and is usually slower to evaluate because it requires recalculating the forward solutions to perform the backward pass. A third way of backpropagating through NDEs is given by *algebraically reversible solvers*, offering both memory and accuracy efficiency. We refer to [28] for further details.

**Differential equations with events**   Many systems are not adequately modelled by continuous differential equations because they experience jump discontinuities triggered by the internal state of the system. Examples include a bouncing ball or spiking neurons. Such systems are often referred to as *(stochastic) hybrid systems* [21, 37]. When the differential equation is an ODE, there is a rich literature on sensitivity analysis aimed at computing derivatives using the implicit function theorem [11, 12]. If, additionally, the vector fields describing the hybrid system are neural networks, [6] show that NODEs solved up until first event time can be implemented as autodifferentiable blocks and [25] derive the corresponding adjoint equations. Nonetheless, none of these works cover the more general setting of SDEs. The only work, we are familiar with, dealing with sensitivity analysis in this setting is [47], although focused on the problem of optimal control.

**Training techniques for SNNs**   Roughly speaking, these works can be divided into two strands. The first, usually referred to as *backpropagation through time* (BPTT), starts with a Euler approximation of the SNN and does backpropagation by unrolling the computational graph over time; it then uses surrogate gradients as smooth approximations of the gradients of the non-differentiable terms. [58, 46, 40]. This approach is essentially analogous to discretise-then-optimise where the backward pass uses custom gradients for the non-differentiable terms. The second strand computes exact gradients of the spike times using the implicit function theorem. These results are equivalent to optimise-then-discretise and can be used to define adjoint equations as in [56] or to derive forward sensitivities [34]. However, we note that, unless solution trajectories and spike times of the SNN are computed exactly, neither method provides the actual gradients of the implemented solver. Furthermore, the BPTT surrogate gradient approach only covers the Euler approximation whereas many auto-differentiable differential equation solvers are available nowadays, e.g. in `diffrax`. Finally, there is a lot of interest in developing bioplausible learning algorithms where weights can be updated locally and in an online fashion. Notable advances in this direction include [2, 57]. To the best of our knowledge, none of these works cover the case of stochastic SNNs where the neuronal dynamics are modeled as SDEs instead of ODEs.

## 3   Stochastic spiking neural networks as Event SDEs

We shall in this paper be concerned with SDEs where solution trajectories experience jumps triggered by implicitly defined events, dubbed *Event SDEs*. The prototypical example that we come back to throughout is the SNN model composed of SLIF neurons. Here the randomness appears both in the inter-spike dynamics as well as in the firing mechanism. To motivate the general definitions and concepts we start with an informal introduction of SSNNs.

### 3.1   Stochastic spiking neural networks

To achieve a more bioplausible model of neuronal behaviour, one can extend the simple deterministic LIF model by adding two types of noise: a diffusion term in the differential equation describing inter-spike behaviour [33] and stochastic firing [50, 27]. That is, the potential is modelled by eq. (1). Instead of firing exactly when the membrane potential hits a set threshold, we model the spike times (event times) by an inhomogenous Poisson process with intensity $\lambda : \mathbb{R}^e \to \mathbb{R}_+$ which is assumed to be bounded by some constant $C > 0$. This can be phrased as an Event SDE (note that this is essentially the reparameterisation trick) by introducing the additional state variable $s_t$ satisfying

$$ds_t = \lambda(v_{t-})dt, \quad s_0 = \log u$$

where $u \sim \text{Unif}(0, 1)$. The neuron spikes whenever $s_t$ hits 0 from below at which point the membrane potential is reset to a resting level and we sample a new initial condition for $s_t$. We can denote this first spike time by $\tau_1$ and repeat the procedure to generate a sequence of spike times $\tau_1 < \tau_2 < ...$ In practice, we reinitialize $s_t$ at $\log u - \alpha$ for some $\alpha > 0$. It can then be shown that

$$\mathbb{P}\left(t < \tau_{n+1} | \mathcal{F}_{\tau_n}\right) = \min\left\{1, \exp\left(\alpha - \int_{\tau_n}^t \lambda(v_{t-})dt\right)\right\} \quad \text{for } t \in [\tau_n, \tau_{n+1}).$$

It follows that $\tau_{n+1} - \tau_n \geq \alpha/C$ a.s., i.e. $\alpha$ controls the refractory period after spikes, a large value indicating a long resting period.

We can then build a SSNN by connecting such SLIF neurons in a network. In particular, apart from the membrane potential, we now also model the input current of each neuron as affected by the other neurons in the network. Let $K \geq 1$ denote the total number of neurons. We model neuron $k \in [K]$ be the three dimensional vector $y^k = (v^k, i^k, s^k)$ the dynamic of which in between spikes is given by

$$dv_t^k = \mu_1\left(i_t^k - v_t^k\right)dt + \sigma_1 dB_t^k, \quad di_t^k = -\mu_2 i_t^k dt + \sigma_2 dB_t^k, \quad ds_t^k = \lambda(v_t^k; \xi)dt, \quad (2)$$

where $B^k$ is a standard two-dimensional Brownian motion, $\sigma = (\sigma_1, \sigma_2) \in \mathbb{R}^{2 \times 2}$, $\mu = (\mu_1, \mu_2) \in \mathbb{R}^2$, and $\lambda(\cdot; \xi) : \mathbb{R} \to \mathbb{R}_+$ is an intensity function. As before, neuron $k$ fires (or spikes) whenever $s^k$ hits zero from below. Apart from resetting the membrane potential, this event also causes spikes to propagate through the network in a such a way that a spike in neuron $k$ will increment the input current of neuron $j$ by $w_{kj}$. Here $w \in \mathbb{R}^{K \times K}$ is a matrix of weights representing the synaptic weights in the neural network. If one is only interested in specific network architectures such as, e.g., feed-forward, this can be achieved by fixing the appropriate entries in $w$ at 0.

As presented here, there is no way to model information coming into the network. But this would only require a minor change. Indeed, by adding a suitable control term to eq. (2) we can model all relevant scenarios. Since this does not change the theory in any meaningful way (the general theory in Appendix B covers RDEs so an extra smooth control is no issue), we only discuss the more simple model given without any additional input currents.

## 3.2 Model definition

**Definition 3.1** (Event SDE). *Let $N \in \mathbb{N}$ be the number of events. Let $y_0 \in \mathbb{R}^e$ be an initial condition. Let $\mu : \mathbb{R}^e \to \mathbb{R}^e$ and $\sigma : \mathbb{R}^e \to \mathbb{R}^{e \times d}$ be the drift and diffusion vector fields. Let $\mathcal{E} : \mathbb{R}^e \to \mathbb{R}$ and $\mathcal{T} : \mathbb{R}^e \to \mathbb{R}^e$ be an event and transition function respectively. We say that $\left(y, (\tau_n)_{n=1}^N\right)$ is a solution to the Event SDE parameterised by $(y_0, \mu, \sigma, \mathcal{E}, \mathcal{T}, N)$ if $y_T = y_T^N$,*

$$y_t = \sum_{n=0}^N y_t^n \mathbf{1}_{[\tau_n, \tau_{n+1})}(t), \quad \tau_n = \inf\left\{t > \tau_{n-1} : \mathcal{E}(y_t^{n-1}) = 0\right\}, \quad (3)$$

*with $\mathcal{E}(y_{\tau_n}^n) \neq 0$ and*

$$dy_t^0 = \mu(y_t^0)dt + \sigma(y_t^0)dB_t, \quad \text{started at } y_0^0 = y_0, \quad (4)$$

$$dy_t^n = \mu(y_t^n)dt + \sigma(y_t^n)dB_t, \quad \text{started at } y_{\tau_n}^n = \mathcal{T}\left(y_{\tau_n}^{n-1}\right), \quad (5)$$

*where $B_t$ is a d-dimensional Brownian motion and (4), (5) are Stratonovich SDEs.*

In words, we initialize the system at $y_0$, evolve it using (4) until the first time $\tau_1$ at which an event happens $\mathcal{E}(y_{\tau_1}^0) = 0$. We then transition the system according to $y_{\tau_1}^1 = \mathcal{T}\left(y_{\tau_1-}^0\right)$ and evolve it according to (5) until the next event is triggered. We note that Definition 3.1 can be generalised to multiple event and transition functions. Also, the transition function can be randomised by allowing it to have an extra argument $u \sim \text{Unif}([0, 1])$. As part of the definition we require that there are only finitely many events and that an event is not immediately triggered upon transitioning.

Existence of strong solutions to Event SDEs driven by continuous semimartingales has been studied in [31, Theorem 5.2] and [32]. Under sufficient regularity of $\mu$ and $\sigma$, a unique solution to (4) exists. We need the following additional assumptions:

*Assumption* 3.1. There exists $c > 0$ such that for all $s \in (0, T)$ and $a \in \text{im } \mathcal{T}$ it holds that $\inf\{t > s : \mathcal{E}(y_t) = 0\} > c$ where $y_t$ is the solution to 4 started at $y_s = a$

*Assumption* 3.2. It holds that $\mathcal{T}(\ker \mathcal{E}) \cap \mathcal{E} = \emptyset$.

Assumptions 3.1 and 3.2 simply ensure that an event cannot be triggered immediately upon transitioning. This holds in most settings of interest. For example, for the usual deterministic LIF neuron im $\mathcal{T} = 0$ and ker $\mathcal{E} = 1$ and the duration of the refractory period is directly linked to $c$ in Assumption 3.1.

**Theorem 3.1** (Theorem 5.2, [31]). *Under Assumptions 3.1-3.2 and with $\mu \in \mathrm{Lip}^1$ and $\sigma \in \mathrm{Lip}^\gamma$ for $\gamma > 2$, there exists a unique solution $(y, (\tau_n)_{n=1}^N)$ to the Event SDE of Definition 3.1.*

The definitions and results of this section can be extended to differential equations driven by random rough paths, and in particular, to cases where the driving noise exhibits jumps. In the latter case, it is important to note that the resulting Event SDE will exhibit two types of jumps: the ones given apriori by the driving noise and the ones that are implicitly defined through the solution (what we call *events*). In fact, we develop the main theory of Event RDEs in Appendix A in the more general setting of RDEs driven by càdlàg rough paths. The rough path formalism enables a unified treatment of differential equations driven by noise signals of arbitrary (ir-)regularity, and makes all proofs simple and systematic. In particular, it allows us to handle cases where the diffusion term is driven by a finite activity Lévy process (e.g, a homogeneous Poisson processes highly relevant in the context of SNNs).

## 3.3 Backpropagation

We are interested in optimizing a continuously differentiable loss function $L$ whose input is the solution of a parameterised Event SDE. As for Neural ODEs, the vector fields, $\mu, \sigma$, and the event and transition functions $\mathcal{E}, \mathcal{T}$, might depend on some learnable parameters $\theta$. We can move the parameters $\theta$ of the Event RDE inside the initial condition $y_0$ by augmenting the dynamics with the additional state variable $\theta_t$ satisfying $d\theta_t = 0$ and $\theta_0 = \theta$. Thus, as long as we can compute gradients with respect to $y_0$, these will include gradients with respect to such parameters. We then require the gradients $\partial_{y_0} L$, if they exist. For this, we need to be able to compute the Jacobians $\partial y_t^n := \partial_{y_0} y_t^n$ of the inter-event flows associated to the dynamics of $y_t^n$ and the derivatives $\partial \tau_n := \partial_{y_0} \tau_n$. We assume that the event and transition functions $\mathcal{E}$ and $\mathcal{T}$ are continuously differentiable.

Apriori, it is not clear under what conditions such quantities exist and even less how to compute them. This shall be the focus of the present section. We will need the following running assumptions.

*Assumption* 3.3. $\sigma(\mathcal{T}(y)) - \nabla \mathcal{T}(y)\sigma(y) = 0$ for all $y \in \ker \mathcal{E}$.

*Assumption* 3.4. $\nabla \mathcal{E}(y)\sigma(y) = 0$ for all $y \in \ker \mathcal{E}$.

*Assumption* 3.5. $\nabla \mathcal{E}(y)\mu(y) \neq 0$ for all $y \in \ker \mathcal{E}$.

Assumption 3.4 and 3.5 ensure that the event times are differentiable. Intuitively, they state that the event condition is hit only by the drift part of the solution. Assumption 3.4 holds for example if the event functions depend only on a smooth part of the system. Assumption 3.3 is what allows us to differentiate through the event transitions.

**Theorem 3.2.** *Let Assumptions 3.1-3.5 be satisfied and $(y, (\tau_n)_{n=1}^N)$ the solution to the Event SDE parameterized by $(y_0, \mu, \sigma, \mathcal{E}, \mathcal{T}, N)$. Then, almost surely, for any $n \in [N]$, the derivatives $\partial \tau_n$ and the Jacobians $\partial y_t^n$ exist and admit the following recursive expressions*

$$\partial \tau_n = -\frac{\nabla \mathcal{E}(y_{\tau_n}^{n-1})\partial y_{\tau_n}^{n-1}}{\nabla \mathcal{E}(y_{\tau_n}^{n-1})\mu(y_{\tau_n}^{n-1})} \tag{6}$$

$$\partial y_t^n = (\partial_{y_{\tau_n}^n} y_t^n) \left[ \nabla \mathcal{T}(y_{\tau_n}^{n-1})\partial y_{\tau_n}^{n-1} - \left( \mu(y_{\tau_n}^n) - \nabla \mathcal{T}(y_{\tau_n}^{n-1})\mu(y_{\tau_n}^{n-1}) \right) \partial \tau_n \right]. \tag{7}$$

*where $\partial y_t^n$ and $\partial \tau_n$ are the total derivatives of $y_n^t$ and $\tau_n$ with respect to the initial condition $y_0$, $\partial_{y_{\tau_n}^n} y_t^n$ denotes the partial derivative of the flow map of eq. (5) with respect to its initial condition, and $\nabla \mathcal{T} \in \mathbb{R}^{e \times e}$ and $\nabla \mathcal{E} \in \mathbb{R}^{1 \times e}$ are the Jacobians of $\mathcal{T}$ and $\mathcal{E}$.*

*Remark* 3.1. If the diffusion term is absent we recover the gradients in [6]. In this case, the assumptions of the theorem are trivially satisfied. Note however, that the result, as stated here, is slightly different since we are considering repeated events.

*Remark* 3.2. The recursive nature of (6) - (7) suggest a way to update gradients in an online fashion by computing the forward sensitivity along with the state of the Event SDE. In traditional machine

learning applications (e.g. NDEs) forward mode automatic differentiation is usually avoided due to the fact that the output dimension tends to be orders of magnitude smaller than the number of parameters [28]. However, for (S)SNNs this issue can be partly avoided as discussed in Section 4.4.

Returning now to the SSNN model introduced in Section 3.1 we find that it is an Event SDE with $K$ different event functions given by $\mathcal{E}_k(y) = s^k$ and corresponding transition functions given by

$$\mathcal{T}_k(y) = \left(\mathcal{T}_k^1(y^1), \ldots, \mathcal{T}_k^K(y^K)\right)$$

where $\mathcal{T}_k^j(y^j) = (v^j, i^j + w_{kj}, s^j)$ if $j \neq k$ and $\mathcal{T}_k^k(y^k) = (v^k - v_{reset}, i^k, \log u - \alpha)$ where $v_{reset} > 0$ is a constant determining by what amount the membrane potential is reset. The addition of the constant $\alpha > 0$ controlling the refractory period ensures that Assumption 3.2 and 3.2 are satisfied. Stochastic firing smooths out the event triggering so that Assumption 3.5 and 3.4 hold. Finally, one can check that the combination of constant diffusion terms and the given transition functions satisfies Assumptions 3.3. Note that setting $v_t^k$ exactly to 0 upon spiking would break Assumption 3.3. If one is interested in such a resetting mechanism it suffices to pick a diffusion term $\sigma_1(y^k)$ that satisfies $\sigma(0) = 0$. To sum up, solutions (in the sense of Def. 3.1) of the SSNNs exist and are unique. In addition, the trajectories and spike times are almost surely differentiable satisfying (6) and (7).

### 3.4 Numerical solvers

Theorem 3.2 gives an expression for the gradients of the event times as well as the Event SDE solution. In practice, analytical expressions for gradients are often not available and one has to resort to numerical solvers. Three solutions suggest themselves:

1. There are multiple autodifferentiable differential equation solvers (such as `diffrax` [28]) that provide differentiable numerical approximations of the flows $\partial_{y_{\tau_n}^n} y_t^n$. We shall write $\mathrm{SDESolve}(y_0, \mu, \sigma, s, t)$ for a generic choice of such a solver. Furthermore, if $\mathrm{RootFind}(y_0, f)$ is a differentiable root finding algorithm (here $f : (y, t) \mapsto \mathbb{R}$ should be differentiable in both arguments and $\mathrm{RootFind}(y_0, f)$ returns $t^* \in \mathbb{R}$ such that $f(y_0, t^*) = 0$), then we can define a differentiable map $E : y_0 \mapsto y^*$ by

   $$t^* = \mathrm{RootFind}\left(y_0, \mathcal{E}(\mathrm{SDESolve}\left(\cdot, \mu, \sigma, s, \cdot\right))\right), \quad y^* = \mathrm{SDESolve}\left(y_0, \mu, \sigma, s, t^*\right).$$

   Consequently, $\mathrm{EventSDESolve}(y_0, \mu, \sigma, \mathcal{E}, \mathcal{T}, N)$ can be implemented as subsequent compositions of $\mathcal{T} \circ E$ (see Algorithm 1). This is a discretise-then-optimise approach [28].

2. Alternatively, one can use the formulas (6) and (7) directly as a replacement of the derivatives. This is the approach taken in e.g. [6]. To be precise, one would replace all the derivatives of the flow map (terms of the sort $\partial_{y_{\tau_n}^n} y_t^n$) with the derivatives of the given numerical solver. This approach is a solution between discretise-then-optimise and optimise-then-discretise.

3. Finally, one could apply the adjoint method (or optimise-then-discretise) as done for deterministic SNNs in [56] by deriving the adjoint equations. These adjoint equations define another SDE with jumps which is solved backwards in time. Between events the dynamics are exactly as in the continuous case so one just needs to specify the jumps of the adjoint process. This can be done by referring to (6) and (7).

*Remark* 3.3. One thing to be careful of with the discretise-then-optimise approach is that the SDE solver will compute time derivatives in the backward pass, although the modelled process is not time differentiable. Assumptions 3.4 and 3.3 should in principle guarantee that these derivatives cancel out (see Appendix B), yet this might not necessarily happen at the level of the numerical solver because of precision issues. This is essentially due to the fact that approximate solutions provided by numerical solvers are in general not *flows*. Thus, when the path driving the diffusion term is very irregular, the gradients can become unstable. In practice we found this could be fixed by setting the gradient with respect to time of the driving Brownian motion to 0 and picking a step size sufficiently small.

*Remark* 3.4. In the context of SNNs, Algorithm 1 is actually a version of *exact backpropagation through time* (BPTT) of the unrolled numerical solution. Contrary to popular belief, this illustrates that one can compute exact gradients of numerical approximations of SNNs without the need to resort to surrogate gradient functions. Of course, this does not alleviate the so-called *dead neuron problem*. However, this ceases to be a problem when stochastic firing is introduced. In fact, surrogate gradients can be related to stochastic firing mechanisms and expected gradients [20].

*Remark* 3.5. One the one hand, the EventSDESolve algorithm as presented here scales poorly in the number of events since it requires doing a full SDESolve and an additional RootFind each time an event occurs. This problem becomes especially prevalent for SSNNs with a large number of neurons since in this case an event is triggered every time a single neuron spikes and the inter-spike SDE that needs to be solved is high-dimensional. On the other hand, there are multiple ways to mitigate this issue. Firstly, one could relax the root-finding step and simply trigger a spike as soon as $e \geq 0$ and take this as the spike time. For the backward pass one could then solve the adjoint equations (for which you need need to store the spike times in the forward pass). The resulting algorithm would be similar to the one derived in [55] for deterministic SNNs. Secondly, for special architectures such as a feed-forward network, given the spikes from the previous layer, one could solve the EventSDE for each neuron in the current layer independently of all other neurons. This would imply that a forward (or backward) pass of the entire SSNN scales as $O(KS)$ where $S$ is the cost of the forward (or backward) pass of a single neuron and $K$ is the number of neurons.

---

**Algorithm 1** EventSDESolve

    **Input** $y_0, \mu, \sigma, \mathcal{E}, \mathcal{T}, N, t_0, \Delta t, T$
1:   $y \leftarrow y_0$
2:   $n \leftarrow 0$
3:   $e \leftarrow \mathcal{E}(y)$
4:   **while** $n < N$ **and** $t_0 < T$ **do**
5:      **while** $e < 0$ **do**                                      ▷ We assume for simplicity that $e \leq 0$
6:         $y_0 \leftarrow y$
7:         $y \leftarrow \text{SDESolveStep}(y_0, \mu, \sigma, t_0, \Delta t)$
8:         $t_0 \leftarrow t_0 + \Delta t$
9:         $e \leftarrow \mathcal{E}(y)$                                           ▷ Update value of event function
10:      **end while**
11:      $t_{n+1}^* \leftarrow \text{RootFind}\left(y_0, \mathcal{E}(\text{SDESolveStep}(\cdot, \mu, \sigma, t_0 - \Delta t, \cdot))\right)$      ▷ Find exact event time
12:      $y_{n+1}^* \leftarrow \text{SDESolveStep}(y_0, \mu, \sigma, t_0 - \Delta_t, t_{n+1}^*)$      ▷ Compute state at event time
13:      $y \leftarrow \mathcal{T}(y_{n+1}^*)$                                            ▷ Apply transition function
14:      $n \leftarrow n + 1$
15: **end while**
    **Return** $(t_n^*)_{n \leq N}, y$

---

## 4 Training stochastic spiking neural networks

### 4.1 A loss function based on signature kernels for càdlàg paths

To train SSNNs we will adopt a similar technique as in [23], where the authors propose to train NSDEs non-adversarially using a class of maximum mean discrepancies (MMD) endowed with signature kernels [52] indexed on spaces of continuous paths as discriminators. However, as we mentioned in the introduction, classical signature kernels are not directly applicable to the setting of SSNNs as the solution trajectories not continuous. To remedy this issue, in Appendix C, we generalise signature kernels to *Marcus signature kernels* indexed on discontinuous (or càdlàg) paths. We note that our numerical experiments only concern learning from spike trains, which are càdlàg paths of bounded variation. Yet, the Marcus signature kernel defined in Appendix C can handle more general càdlàg rough paths.

The main idea goes as follows. If $x$ is a càdlàg path, one can define the *Marcus signature* $S(x)$ in the spirit of Marcus SDEs [42, 43] as the signature of the *Marcus interpolation* of $x$. The general construction is given in Appendix A. The *Marcus signature kernel* is defined as the inner product $k(x, y) = \langle S(x), S(y) \rangle$ of Marcus signatures $S(x), S(y)$ of two càdlàg paths $x, y$. As stated in the first part of Theorem C.1, this kernel is characteristic on regular Borel measures supported on compact sets of càdlàg paths. In particular, this implies that the resulting *maximum mean discrepancy* (MMD)

$$d_k(\mu, \nu)^2 = \mathbb{E}_{x, x' \sim \mu} k(x, x') - 2\mathbb{E}_{x, y \sim \mu \times \nu} k(x, x') + \mathbb{E}_{y, y' \sim \nu} k(y, y')$$

satisfies the property $d_k(\mu, \nu)^2 = 0 \iff \mu = \nu$ for any two compactly supported measures $\mu, \nu$.

Nonetheless, characteristicness ceases to hold when one considers measures on càdlàg paths that are not compactly supported. In [8] the authors address this issue for continuous paths by using the

so-called *robust signature*. They introduce a *tensor normalization* $\Lambda$ ensuring that the range of the robust signature $\Lambda \circ S$ remains bounded. The *robust signature kernel* is then defined as the inner product $k_\Lambda(x, y) = \langle \Lambda \circ S(x), \Lambda \circ S(y) \rangle$. This normalization can be applied analogously to the *Marcus signature* resulting in a *robust Marcus signature kernel*. In the second part of Theorem C.1, we prove characteristicness of $k_\Lambda$ for possibly non-compactly supported Borel measures on càdlàg paths. The resulting MMD is denoted by $d_{k_\Lambda}$.

There are several ways of evaluating signature kernels. The most naive is to simply truncate the signatures at some finite level and then take their inner product. Another amounts to solve a path-dependent wave equation [52]. Our experiments are compatible with both of these methods.

Given a collection of observed càdlàg trajectories $\{x^i\}_{i=1}^m \sim \mu^{\text{true}}$ sampled from an underlying unknown target measure $\mu^{\text{true}}$, we can train an Event SDE by matching the generated càdlàg trajectories $\{y^i\}_{i=1}^n \sim \mu^\theta$ using an unbiased empirical estimator of $d_k$ (or $d_{k_\Lambda}$), i.e. minimising over the parameters $\theta$ of the Event SDE the following loss function

$$\mathcal{L} = \frac{1}{m(m-1)} \sum_{j \neq i} k(x^i, x^j) - \frac{2}{mn} \sum_{i,j} k(x^i, y^j) + \frac{1}{n(n-1)} \sum_{j \neq i} k(y^i, y^j).$$

In the context of SSNNs, the observed and generated trajectories $x^i$'s and $y^i$'s correspond to spike trains, which are càdlàg paths of bounded variation.

## 4.2 Input current estimation

The first example is the simple problem of estimating the constant input current $c > 0$ based on a sample of spike trains in the single SLIF neuron model,

$$dv_t = \mu(c - v_t)dt + \sigma dB_t, \quad ds_t = \lambda(v_t)dt,$$

where $\lambda(v) = \exp(5(v-1)$, $\mu = 15$ and $\sigma$ varies. Throughout we fix the true $c = 1.5$ and set $v_{reset} = 1.4$ and $\alpha = 0.03$. We run stochastic gradient descent for 1500 steps for two choices of the diffusion constant $\sigma$. The loss function is the signature kernel MMD between a simulated batch and the sample of spike trains.[3]. The test loss is the mean absolute error between the first three average spike times. Results are given in Fig. 1. For additional details regarding the experiments, we refer to Appendix E.

In all cases backpropagation through Algorithm 1 is able to learn the underlying input current after around 600 steps up to a small estimation error. In particular, the convergence is fastest for the largest sample size and the true $c$ is recovered for both levels of noise.

## 4.3 Synaptic weight estimation

Next we consider the problem of estimating the weight matrix in a feed-forward SSNN with input dimension 4, 1 hidden layer of dimension 16, and output dimension 2. The rest of the parameters are fixed throughout. We run stochastic gradient descent for 1500 steps with a batch size of 128 and for a sample size of $256, 512$, and $1024$ respectively. Learning rate is decreased from $0.003$ to $0.001$ after 1000 steps. The results are given in Fig. 2 in Appendix E. For a sample size of $512$ and $1024$ we are able to reach a test loss of practically 0, that is, samples from the learned model and the underlying model are more or less indistinguishable. Also, in all cases the estimated weight matrix approaches the true weight matrix. Interestingly, for the largest sample size, the model reaches the same test loss as the model trained on a sample size of $512$, but their estimated weight matrices differ significantly.

## 4.4 Online learning

In the case of SSNNs, equations (6)-(7) lead to a formula for the forward sensitivity where any propagation of gradients between neurons only happens at spike times and only between connected neurons (see Proposition D.1). Since the forward sensitivities are computed forward in time together with the solution of the SNN, gradients can be updated online as new data appears. As a result, between spikes of pre-synaptic neurons, we can update the gradient flow of the membrane potential

---

[3]For simplicity we only compute an approximation of the true MMD by truncating the signatures at depth 3 and taking the average across the batch/sample size.

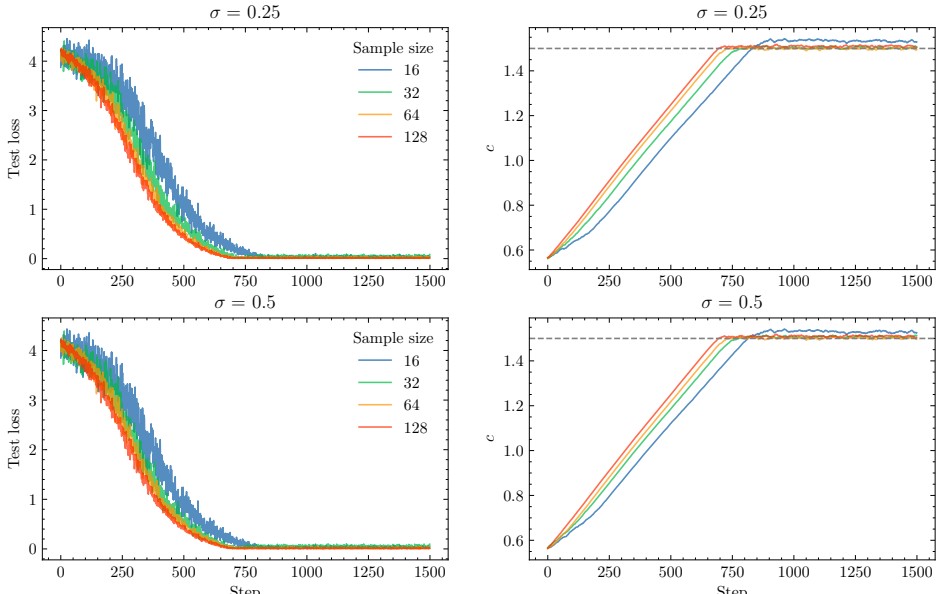

Figure 1: Test loss and $c$ estimate across four sample sizes and for two levels of noise $\sigma$. On the left: MAE for the three first average spike times on a hold out test set. On the right: estimated value of $c$ at the current step.

and input current of each neuron using information exclusively from that neuron. For general network structures and loss functions, however, this implies that each neuron needs to store on the order of $K^2$ gradient flows (one for each weight in the network).

On the other hand, if the adjacency matrix of the weight matrix forms a directed acyclic graph (DAG), three-factor Hebbian learning rules like those in [57, 2] are easily derived from Proposition D.1. For simplicity, consider the SNN consisting of deterministic LIF neurons and let $N_t^k$ denote the spike train of neuron $k$, i.e., $N_t^k$ is càdlàg path equal to the number of spikes of neuron $k$ at time $t$. We let $\tau^k(t)$ (or $\tau^k$ for short) denote the last spike of neuron $k$ before time $t$. We shall assume that the instantaneous loss function $L_t$ depends only on the most recent spike times $\tau^1, \ldots, \tau^K$. Then,

$$\partial_{w_{jk}} L_t = \partial_{\tau^k} L_t \frac{a_{\tau^k}^{jk}}{\mu_1(v_{\tau^k}^k - i_{\tau^k}^k)}$$

where $a_t^{jk}$ is the *eligibility trace* and the first term can be viewed as a *global modulator*, that is, a top-down learning signal propagating the error from the output neurons.[4] The eligibility trace satisfies

$$da_t^{jk} = \mu_1 \left( b_t^{jk} - a_t^{jk} \right) dt + \frac{v_{reset} a_t^{jk}}{\mu_1(i_t^k - v_t^k)} dN_t^k, \quad db_t^{jk} = -\mu_2 b_t^{jk} + dN_t^j,$$

where the $dN$ terms are to be understood in the Riemann-Stieltjes sense. In other words, the eligibility trace can be updated exclusively from the activity of the pre- and post-synaptic neurons. We note the similarity to the results derived in [2] only our result gives the exact gradients with no need to introduce surrogate gradient functions. A similar equation for deterministic SNNs was derived in [49] (see, in particular, Chapter 5). For general network structures one can use the eligibility traces as proxies for the the true derivatives $\partial_{w_{ij}} \tau^k$.

## 5 Conclusion

We introduced a mathematical framework based on rough path theory to model SSNNs as SDEs exhibiting event discontinuities and driven by càdlàg rough paths. After identifying sufficient

---

[4]Note that in the case of stochastic SNNs this term is not necessarily well-defined since semi-martingales are in general not differentiable wrt. time.

conditions for differentiability of solution trajectories and event times, we obtained a recursive relation for the pathwise gradients in Theorem 3.2, generalising the results presented in [6] and [25] which only deal with the case of ODEs. Next, we introduced Marcus signature kernels as extensions of continuous signature kernels from [52] to càdlàg rough paths and used them to define a general-purpose loss function on the space of càdlàg rough paths to train SSNNs where noise is present in both the spike timing and the network's dynamics. Based on these results, we also provided an end-to-end autodifferentiable solver for SDEs with event discontinuities (Algorithm 1) and made its implementation available as part of the `diffrax` repository. Finally, we discussed how our results lead to bioplausible learning algorithms akin to *e-prop* [2] but in the context of spike time gradients.

The primary objective of the paper was to lay out the theoretical foundations of gradient-based learning with stochastic SNNs. Although we provided an initial implementation, which is well-suited for low dimensional examples, a robust version that scales to a high number of neurons is beyond the scope of the paper. Examples that require a much higher number of neurons than the two examples already discussed will be hard to handle with the discretize-then-optimize approach for the reasons given in Remark 3.5.

We think there are still many interesting research directions left to explore. For instance, it would be of interest to implement the adjoint equations or to use reversible solvers and compare the results. Similarly, since our Algorithm 1 differs from the usual approach with surrogate gradients even in the deterministic setting, questions remain on how these methods compare for training SNNs. Furthermore, it would be interesting to understand to what extent the inclusion of different types of driving noises in the dynamics of SSNNs would be beneficial for learning tasks compared to deterministic SNNs. Finally, it remains to be seen whether the discussion in Section 4.4 could lead to a bio-plausible learning algorithm with comparable performance to state-of-the-art backpropagation methods and implementable on neuromorphic hardware.

**Acknowledgements**. Christian Holberg gratefully acknowledges financial support from Novo Nordisk Foundation through Grant NNF20OC0062958 and from Independent Research Fund Denmark | Natural Sciences through Grant 9040-00215B.

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

## Appendix

The appendix is structured as follows. Section A covers the basic concepts of càdlàg rough paths based on [7] extended with a few of our own definitions and results. It culminates with the definition of Event RDEs which can be viewed as generalizations of Event SDEs. Section B covers the proof of the main result, Theorem 3.2, but in the setting of Event RDEs as well as some preliminary technical lemmas needed for the proof. Section C gives a brief overview of the main concepts in kernel learning and presents our results on Marcus signature kernels along with their proofs. Section D derives the forward sensitivities of a SSNN. Finally, Section E covers all the technical details of the simulation experiments that were not discussed in the main body of the paper.

## A  Càdlàg rough paths

*Marcus integration* developed in [7] preserves the chain rule and thus serves as an analog to Stratonovich integration for semi-martingales with jump discontinuities. In particular, it allows to define a canonical lift under which càdlàg semi-martingales are a.s. geometric rough paths and many of the results from the continuous case, such as universal limit theorems and stability results, carry over under suitably defined metrics. We briefly review some of the important concepts here by following the same setup as in [7].

Let $C([0,T], E)$ and $D([0,T], E)$ be the space of continuous and càdlàg paths respectively on $[0,T]$ with values in a metric space $(E, d)$. For $p \geq 1$, let $C_p([0,T], E)$ and $D_p([0,T], E)$ be the corresponding subspaces of paths with finite $p$-variation. For any $N \geq 1$, Let $G^N(\mathbb{R}^d)$ be the step-$N$ free nilpotent Lie group over $\mathbb{R}^d$ endowed with the Carnot-Carathéodory metric $d$. Let $\Omega_p^C(\mathbb{R}^d) := C_p([0,T], G^{\lfloor p \rfloor}(\mathbb{R}^d))$ and $\Omega_p^D(\mathbb{R}^d) := D_p([0,T], G^{\lfloor p \rfloor}(\mathbb{R}^d))$ be the space of weakly geometric continuous and càdlàg $p$-rough paths respectively with the homogeneous $p$-variation metric

$$d_p(\mathbf{x}, \mathbf{y}) = \max_{1 \leq k \leq \lfloor p \rfloor} \sup_{\mathcal{D} \subset [0,T]} \left( \sum_{\mathcal{D}} d(\mathbf{x}_{t_i, t_{i+1}}, \mathbf{y}_{t_i, t_{i+1}})^{\frac{p}{k}} \right)^{\frac{k}{p}}.$$

Define the *log-linear* path function

$$\phi : G^N(\mathbb{R}^d) \times G^N(\mathbb{R}^d) \to C([0,1], G^N(\mathbb{R}^d))$$
$$(\mathbf{a}, \mathbf{b}) \mapsto \exp((1 - \cdot) \log \mathbf{a} + \cdot \log \mathbf{b}).$$

where $\log$ and $\exp$ are the (truncated) tensor logarithm and exponential maps on $G^N(\mathbb{R}^d)$. If $N = 1$, then $G^N(\mathbb{R}^d) \cong \mathbb{R} \oplus \mathbb{R}^d$ and $\phi(a, b)_t = (1, (1 - t)a + tb)$ is a straight line connecting $a$ to $b$ in unit time. For any $\mathbf{x} \in D([0,T], G^N(\mathbb{R}^d))$ we can construct a continuous path $\hat{\mathbf{x}} \in C([0,T], G^N(\mathbb{R}^d))$ by adding fictitious time and interpolating through the jumps using the log-linear path function according to the following definition.

**Definition A.1** (Marcus interpolation). *Let $N \geq 1$. For $\mathbf{x} \in D([0,T], G^N(\mathbb{R}^d))$, let $\tau_1, \tau_2, \ldots, \tau_m$ be the jump times of $\mathbf{x}$ ordered such that $d(\mathbf{x}_{\tau_1-}, \mathbf{x}_{\tau_1}) \geq d(\mathbf{x}_{\tau_2-}, \mathbf{x}_{\tau_2}) \geq \cdots \geq d(\mathbf{x}_{\tau_m-}, \mathbf{x}_{\tau_m})$, where $0 \leq m \leq \infty$ is the number of jumps. Let $(r_k)$ be a sequence of positive scalars $r_k > 0$ such that $r = \sum_{k=1}^m r_k < +\infty$. Define the discontinuous reparameterisation $\eta : [0, T] \to [0, T + r]$ by*

$$\eta(t) = t + \sum_{k=1}^m r_k \mathbf{1}_{\{\tau_k \leq t\}}.$$

*The Marcus augmentation $\mathbf{x}^M \in C([0, T + r], G^N(\mathbb{R}^d))$ of $\mathbf{x}$ is the path*

$$\mathbf{x}_s^M = \begin{cases} \mathbf{x}_t, & \text{if } s = \eta(t) \text{ for some } t \in [0, T], \\ \phi(\mathbf{x}_{\tau_k-}, \mathbf{x}_{\tau_k})_{(s - \eta(\tau_k-))/r_k}, & \text{if } s \in [\eta(\tau_k-), \eta(\tau_k)) \text{ for } 1 \leq k < m + 1. \end{cases}$$

*The Marcus interpolation $\hat{\mathbf{x}} \in C([0, T], G^N(\mathbb{R}^d))$ of $\mathbf{x}$ is the path $\hat{\mathbf{x}} = \mathbf{x}^M \circ \eta_r$ where $\eta_r(t) = t(T + r)/T$ is a reparameterisation from $[0, T]$ to $[0, T + r]$. We can recover $\mathbf{x}$ from $\hat{\mathbf{x}}$ via $\mathbf{x} = \hat{\mathbf{x}} \circ \eta_{\mathbf{x}}$ by considering the reparameterisation $\eta_{\mathbf{x}} = \eta_r^{-1} \circ \eta$.*

Once the Marcus interpolation is defined we can state what we mean by a solution to a differential equation driven by a geometric càdlàg rough path.

**Definition A.2** (Marcus RDE)**.** *Let* $\mathbf{x} \in \Omega_p^D(\mathbb{R}^d)$ *and* $f = (f_1, \ldots, f_d)$ *be* $\mathrm{Lip}^\gamma$ *vector fields on* $\mathbb{R}^e$ *with* $\gamma > p$. *For an initial condition* $a \in \mathbb{R}^e$, *let* $\hat{y} \in C_p([0,T], \mathbb{R}^e)$ *be the solution to the classical RDE driven by the Marcus interpolation* $\hat{\mathbf{x}} \in \Omega_p^C(\mathbb{R}^d)$

$$d\hat{y}_t = f(\hat{y}_t)d\hat{\mathbf{x}}_t, \quad \hat{y}_0 = a.$$

*Define the solution* $y \in D_p([0,T], \mathbb{R}^e)$ *to the Marcus RDE*

$$dy_t = f(y_t) \diamond d\mathbf{x}_t, \quad y_0 = a \tag{8}$$

*to be* $y = \hat{y} \circ \eta_{\mathbf{x}}$, *where* $\eta_{\mathbf{x}}$ *is the reparameterisation introduced in Definition A.1.*

### A.1 Metrics on the space of càdlàg rough paths

Chevyrev and Friz [7] introduce a metric $\alpha_p$ on $\Omega_p^D(\mathbb{R}^d)$ with respect to which 1) geometric càdlàg rough paths can be approximated with a sequence of continuous paths [7, Section 3.2] and 2) the solution map $(y_0, \mathbf{x}) \mapsto (\mathbf{x}, y)$ of the Marcus RDE (8) is locally Lipschitz continuous [7, Theorem 3.13].

We write $\Lambda$ for the set of increasing bijections from $[0,T]$ to itself. For a $\lambda \in \Lambda$ we let $|\lambda| = \sup_{t \in [0,T]} |\lambda(t) - t|$. We first define the Skorokhod metric as well as a Skorokhod version of the usual $p$-variation metric.

**Definition A.3.** *For* $p \geq 1$ *and* $\mathbf{x}, \mathbf{y} \in D_p([0,T], E)$, *we define*

$$\sigma_\infty(\mathbf{x}, \mathbf{y}) = \inf_{\lambda \in \Lambda} \max \left\{ |\lambda|, \sup_{t \in [0,T]} d((\mathbf{x} \circ \lambda)_t, \mathbf{y}_t) \right\},$$

$$\sigma_p(\mathbf{x}, \mathbf{y}) = \inf_{\lambda \in \Lambda} \max \left\{ |\lambda|, d_p(\mathbf{x} \circ \lambda, \mathbf{y}) \right\}.$$

It turns out that the topology induced by $\sigma_p$ is too strong. In particular, it is not possible to approximate paths with jump discontinuities with a sequence of continuous paths (see Section 3.2 in [7]). For $\mathbf{x} \in \Omega_p^D(\mathbb{R}^d)$ and $f = (f_1, \ldots, f_d)$ a family of vector fields in $\mathrm{Lip}^{\gamma-1}(\mathbb{R}^e)$ with $\gamma > p$, let $\Phi_f(y, s, t; \mathbf{x})$ denote the solution to the Marcus RDE $dy_t = f(y_t) \diamond d\mathbf{x}_t$ initialized at $y_s = y$ and evaluated at time $t$. We define the set

$$J_f = \left\{ ((\mathbf{a}, b), (\mathbf{a}', b')) \mid \mathbf{a}, \mathbf{a}' \in G^{\lfloor p \rfloor}(\mathbb{R}^e), \Phi_f(b, 0, 1; \phi(\mathbf{a}, \mathbf{a}')) = b' \right\}.$$

and, on it, the path function

$$\phi_f((\mathbf{a}, b), (\mathbf{a}', b'))_t = (\phi(\mathbf{a}, \mathbf{a}'), \Phi_f(b, 0, 1; \phi(\mathbf{a}, \mathbf{a}')_t)).$$

Finally, we let $D_p^f([0,T], G^{\lfloor p \rfloor}(\mathbb{R}^d) \times \mathbb{R}^e)$ be the space of càdlàg paths $\mathbf{z} = (\mathbf{x}, y)$ on $G^{\lfloor p \rfloor}(\mathbb{R}^d) \times \mathbb{R}^e$ of bounded $p$-variation such that $(\mathbf{z}_t-, \mathbf{z}_t) \in J_f$ for all jump times $t$ of $\mathbf{z}$. To keep notation simple, we shall write $D_p^f$ when this does not cause any confusion. Naturally, if $y$ is the solution to the Marcus RDE $dy_t = f(y_t) \diamond d\mathbf{x}_t$, we have $(\mathbf{x}, y) \in D_p^f$. For a $\mathbf{z} = (\mathbf{x}, y) \in D_p^f$ we may define the Marcus interpolation by interpolating the jumps using $\phi_f$. Let $\hat{\mathbf{z}}^\delta$ denote this interpolation but with $r_k$ replaced by $\delta r_k$ for $\delta > 0$ and similarly for $\hat{\mathbf{x}}^\delta$ with $\mathbf{x} \in \Omega_p^D(\mathbb{R}^d)$.

**Definition A.4.** *For* $f = (f_1, \ldots, f_d)$ *a family of vector fields in* $\mathrm{Lip}^{\gamma-1}(\mathbb{R}^e)$ *with* $\gamma > p$, *let* $\mathbf{z}, \mathbf{z}' \in D_p^f$ *with* $\mathbf{z} = (\mathbf{x}, y)$ *and* $\mathbf{z}' = (\mathbf{x}', y')$ *and define*

$$\alpha_p(\mathbf{x}, \mathbf{x}') = \lim_{\delta \to 0} \sigma_p(\hat{\mathbf{x}}^\delta, \hat{\mathbf{x}}'^\delta),$$

$$\alpha_p(\mathbf{z}, \mathbf{z}') = \lim_{\delta \to 0} \sigma_p(\hat{\mathbf{z}}^\delta, \hat{\mathbf{z}}'^\delta).$$

*Remark* A.1. It is proven in [7] that in both cases the limit in $\alpha_p$ exists, is independent of the choice of $r_k$, and that it is indeed a metric on $\Omega_p^D(\mathbb{R}^d)$ resp. $D_p^f$.

**Theorem A.1** (Theorem 3.13 + Proposition 3.18, [7])**.** *Let* $f = (f_1, \ldots, f_d)$ *be a family of vector fields in* $\mathrm{Lip}^{\gamma-1}(\mathbb{R}^e)$ *with* $\gamma > p$. *Then,*

1. *The solution map*
$$\mathbb{R}^e \times (\Omega_p^D(\mathbb{R}^d), \alpha_p) \to (D_p^f, \alpha_p)$$
$$(y_0, \mathbf{x}) \mapsto \mathbf{z} = (\mathbf{x}, y)$$
   *of the Marcus RDE $dy_t = f(y_t) \diamond d\mathbf{x}_t$ initialized at $y_0 \in \mathbb{R}^e$ is locally Lipschitz.*

2. *On sets of bounded $p$-variation, the solution map*
$$\mathbb{R}^e \times (\Omega_p^D(\mathbb{R}^d), \sigma_\infty) \to (D_p([0,T], \mathbb{R}^e), \sigma_\infty)$$
$$(y_0, \mathbf{x}) \mapsto y$$
   *of the Marcus RDE $dy_t = f(y_t) \diamond d\mathbf{x}_t$ initialized at $y_0 \in \mathbb{R}^e$ is continuous.*

Now, let $C_0^1(\mathbb{R}^d)$ be the space of absolutely continuous functions on $\mathbb{R}^d$.

**Definition A.5.** *We define the space of geometric càdlàg $p$-rough paths $\Omega_{0,p}^D(\mathbb{R}^d)$ as the closure of $C_0^1(\mathbb{R}^d)$ in $\Omega_p^D(\mathbb{R}^d)$ under the metric $\alpha_p$.*

*Remark* A.2. A càdlàg semi-martingale $x \in D_p([0,T], \mathbb{R}^d)$ can be canonically lifted to a geometric càdlàg $p$-rough path, with $p \in [2,3)$, by enhancing it with its two-fold iterated Marcus integrals, i.e.

$$\mathbf{x}_{s,t} = \left(1, x_{s,t}, \int_{s,t} (x_s - x_u) \otimes \diamond dx_u\right) \in G^2(\mathbb{R}^d)$$

where the integral is defined in a similar spirit to Definition A.2 (see, for example, [18] for more information). The solution to the corresponding Marcus RDE agrees a.s. with the solution to the usual càdlàg Marcus SDE which, in turn, if $x$ has a.s. continuous sample paths, agrees a.s. with the solution to the Stratonovich SDE. See, e.g., Proposition 4.16 in [7].

## A.2 Signature

The extended tensor algebra over $\mathbb{R}^d$ is given by

$$T\left((\mathbb{R}^d)\right) = \prod_{n=0}^\infty (\mathbb{R}^d)^{\otimes n}$$

equipped with the usual addition $+$ and tensor multiplication $\otimes$. An element $\mathbf{a} \in T\left((\mathbb{R}^d)\right)$ is a formal series of tensors $\mathbf{a} = (\mathbf{a}^0, \mathbf{a}^1, \dots)$ such that $\mathbf{a}^n \in (\mathbb{R}^d)^{\otimes n}$. We define the projections $\pi_n : T\left((\mathbb{R}^d)\right) \to (\mathbb{R}^d)^{\otimes n}$ given by $\pi_n(\mathbf{a}) = \mathbf{a}^n$. Let $\tilde{T}((\mathbb{R}^d))$ be the subset of $T((\mathbb{R}^d))$ such that the $\pi_0(\mathbf{a}) = 1$ for all $\mathbf{a} \in \tilde{T}((\mathbb{R}^d))$. Finally, we define the set of group-like elements,

$$G^{(*)} = \left\{ \mathbf{a} \in \tilde{T}\left((\mathbb{R}^d)\right) \mid \pi_n(\mathbf{a}) \in G^N(\mathbb{R}^d) \text{ for all } n \geq 0 \right\}$$

**Definition A.6.** *Let $p \geq 1$ and $\mathbf{x} \in \Omega_p^D(\mathbb{R}^d)$. The signature of $\mathbf{x}$ is the path $S(\mathbf{x}) : [0,T] \mapsto G^{(*)}$ such that, for each $N \geq 0$,*

$$dS(\mathbf{x})_t^N = S(\mathbf{x})_t^N \otimes \diamond d\mathbf{x}_t, \quad S(\mathbf{x})_0^N = \mathbf{1} \in G^N(\mathbb{R}^d). \tag{9}$$

*Remark* A.3. Uniqueness and existence of the signature follow from the continuous analog. Indeed, by definition, (9) is equivalent to a continuous linear RDE.

*Remark* A.4. The signature, as defined here, is also known as the *minimal jump extension of* $\mathbf{x}$ and was first introduced in [15]. It was further explored in [13] where it was also shown that it acts as a universal feature map.

In the continuous case, it is well known that the signature characterizes paths up to *tree-like equivalence*. Two continuous paths $\mathbf{x}, \mathbf{y}$ are said to be tree-like equivalent if there exists a continuous non-negative map $h : [0,T] \to \mathbb{R}_+$ such that $h(0) = h(T)$ and

$$\|\mathbf{x}_{s,t} - \mathbf{y}_{s,t}\| \leq h(s) + h(t) - 2 \inf_{u \in [s,t]} h(u).$$

This can be generalized to càdlàg paths in the following way. We say that two càdlàg paths $\mathbf{x}, \mathbf{y}$ are tree-like equivalent, or $\mathbf{x} \sim_t \mathbf{y}$, if their their Marcus interpolations (see Def. A.1), $\hat{\mathbf{x}}$ and $\hat{\mathbf{y}}$, are tree-like equivalent. It is straightforward to check that this indeed is an equivalence relation on $\Omega_p^D(\mathbb{R}^d)$. Perhaps more interestingly, we obtain the following result. For ease of notation we shall henceforth mean $S(\mathbf{x})_T$ when omitting the subscript from the signature.

**Proposition A.1.** *Let $p \geq 1$. The map $S(\cdot) : \Omega_p^D(\mathbb{R}^d) \to G^{(*)}$ is injective up to tree-like equivalence, i.e., $S(\mathbf{x}) = S(\mathbf{y})$ iff $\mathbf{x} \sim_t \mathbf{y}$.*

*Proof.* The result follows from the continuous case upon realizing that $S(\mathbf{x}) = S(\hat{\mathbf{x}})$ and analogously for $\mathbf{y}$. $\qquad\square$

### A.3 Young pairing

In many cases, given a geometric càdlàg rough path $\mathbf{x} \in \Omega_{0,p}^D(\mathbb{R}^d)$ with $p \in [2,3)$ and a path $h \in D_1([0,T],\mathbb{R}^e)$ of bounded variation one is interested in constructing a new rough path $\mathbf{y} \in \Omega_{0,p}^D(\mathbb{R}^{d+e})$ such that the first level of $\mathbf{y}$ is given by $y = (x,h)$. In the continuous case this can be done by using the level two information $\mathbf{x}^2$ and $\int dh \otimes dh$ to fill in the corresponding terms in $\mathbf{y}^2$ and using the well-defined Young cross-integrals to fill in the rest. The resulting level 2 rough path is called the *Young pairing* of $\mathbf{x}$ and $h$ and we will denote it by $\mathbf{y} = P(\mathbf{x},h)$. The canonical example to keep in the mind is when $h_t = t$, that is, we want to augment the rough path with an added time coordinate (see Def. A.8). In the càdlàg case one needs to be more careful in defining the appropriate Marcus lift.

**Definition A.7** (Definition 3.21 [7]). *Let $\mathbf{x} \in \Omega_{0,p}^D(\mathbb{R}^d)$ with $p \geq 1$ and $h \in D_1([0,T],\mathbb{R}^e)$. Define the path $\mathbf{z} = (\mathbf{x},h)$ and the corresponding Marcus lift $\hat{\mathbf{z}} = (\hat{\mathbf{x}},\hat{h})$. The Young pairing of $\mathbf{x}$ and $h$ is the $p$-rough path $P(\mathbf{x},h) \in \Omega_p^D(\mathbb{R}^{d+e})$ such that*

$$P(\mathbf{x},h) = P(\hat{\mathbf{x}},\hat{h}) \circ \eta_{\mathbf{z}}$$

*where $P(\hat{\mathbf{x}},\hat{h})$ is the usual Young pairing of a continuous rough path and a continuous bounded variation path (see Def. 9.27 in [17]).*

We can then construct the time augmented rough path as the rough path obtained by the Young pairing with the simple continuous bounded variation path $h_t = t$. It turns out that this pairing is continuous as a map from $\Omega_{0,p}^D(\mathbb{R}^d)$ to $\Omega_{0,p}^D(\mathbb{R}^{d+1})$.

**Definition A.8.** *Let $\mathbf{x} \in \Omega_{0,p}^D(\mathbb{R}^d)$. The time augmented version of $\mathbf{x}$ is the unique rough path $\tilde{\mathbf{x}} \in \Omega_{0,p}^D(\mathbb{R}^{d+1})$ obtained by the Young pairing $P(\mathbf{x},h)$ of $\mathbf{x}$ with the continuous bounded variation path $h_t = t$.*

**Proposition A.2.** *Let $p \in [1,3)$. Then, the map $\mathbf{x} \mapsto \tilde{\mathbf{x}}$ is continuous and injective as a map from $\Omega_{0,p}^D(\mathbb{R}^d)$ to $\Omega_{0,p}^D(\mathbb{R}^{d+1})$.*

*Proof.* Let $\mathcal{X} = \Omega_{0,p}^D(\mathbb{R}^d)$ be a metric space when equipped with $\alpha_p$. Fix $\mathbf{x} \in \mathcal{X}$ and let $x^n$ be a sequence of absolutely continuous paths converging in $\mathcal{X}$ to $\mathbf{x}$. We shall first show that $\tilde{x}^n$ then converges to $\tilde{\mathbf{x}}$. Since $x^n$ does not have any jumps and any reparameterisation of $x^n$ is still absolutely continuous, we may assume that

$$\alpha_p(\mathbf{x},x^n) = \lim_{\delta \to 0} d_p(\hat{\mathbf{x}}^\delta, x^n) \to 0$$

for $n \to \infty$. Define $\mathbf{z} = (\mathbf{x},h)$ and $\hat{\mathbf{z}}^d = (\hat{\mathbf{x}}^\delta, \hat{h}^\delta)$ the Marcus interpolation with $\eta_{\mathbf{x},\delta}$ the reparameterisation such that $\mathbf{z} = \hat{\mathbf{z}}^\delta \circ \eta_{\mathbf{x},\delta}$. Furthermore, let $P(\mathbf{x},h)$ be the Young pairing of $\mathbf{x}$ and $h$. By definition,

$$
\begin{aligned}
\alpha_p\left(P(\mathbf{x},h), P(x^n,h)\right) &= \lim_{\delta \to 0} \sigma_p\left(P(\hat{\mathbf{x}}^\delta,\hat{h}^\delta), P(x^n,h)\right) \\
&\leq \lim_{\delta \to 0} d_p\left(P(\hat{\mathbf{x}}^\delta,\hat{h}^\delta), P(x^n,h)\right) \\
&\leq \lim_{\delta \to 0} C\left(d_p(\hat{\mathbf{x}}^\delta, x^n) + d_1(\hat{h}^\delta, h)\right) \to 0
\end{aligned}
$$

for $n \to \infty$ where $C$ is just some generic constant depending only on $p$. The last inequality follows from 9.32 in [17]. Thus, if $\mathbf{y} \in \mathcal{X}$ is such that $\alpha_p(\mathbf{x},\mathbf{y}) < \epsilon$, we can choose another sequence $y^n$ of absolutely continuous paths and $N \geq 1$ large enough so that

$$\alpha_p\left(P(\mathbf{x},h), P(\mathbf{y},h)\right) \leq 2\epsilon + \alpha_p(P(x^n,h), P(y^n,h)),$$

$$\alpha_p(x^n, y^n) \leq 2\epsilon$$

for all $n \geq N$. By Remark 3.6 in [7], we then have that, up to choosing a large $N$, $d_p(x^n, y^n) \leq \epsilon$ for all $n \geq N$ and therefore, once more appealing to Theorem 9.32 in [17],

$$\alpha_p\left(P(x^n, h), P(y^n, h)\right) \leq 2C\epsilon.$$

In conclusion, $\alpha_p\left(P(\mathbf{x}, h), P(\mathbf{y}, h)\right) \leq 2(1 + C)\epsilon$ which proves the result.

Injectivity follows from [13]. □

### A.4 Event RDEs

The results of Section 3.3 hold in more generality. In fact, we can define Event RDEs similar to Definition 3.1 where the inter-event dynamics are given by Marcus RDEs driven by càdlàg rough paths. Utilizing the correspondence between solutions to Marcus RDEs and Marcus SDEs, it then follows that the results in the main body of the paper are a special case of the results given below.

**Definition A.9** (Event RDE). *Let $p \geq 1$ and $N \in \mathbb{N}$ be the number of events. Let $\mathbf{x} \in \Omega_p^D(\mathbb{R}^d)$ and $f = (f_1, \ldots, f_d)$ be a family of $\mathrm{Lip}^\gamma$ on $\mathbb{R}^e$ with $\gamma > p$. Let $\mathcal{E} : \mathbb{R}^e \to \mathbb{R}$ and $\mathcal{T} : \mathbb{R}^e \to \mathbb{R}^e$ be an event and transition function respectively. We say that $\left(y, (\tau_n)_{n=1}^N\right)$ is a solution to the Event RDE parameterised by $(y_0, \mathbf{x}, f, \mathcal{E}, \mathcal{T}, N)$ if $y_T = y_T^N$,*

$$y_t = \sum_{n=0}^N y_t^n \mathbf{1}_{[\tau_n, \tau_{n+1})}(t), \quad \tau_n = \inf\left\{t > \tau_{n-1} : \mathcal{E}(y_{t-}^{n-1}) = 0\right\}, \tag{10}$$

*with $\mathcal{E}(y_{\tau_n}^n) \neq 0$ and*

$$dy_t^0 = f(y_t^0) \diamond d\mathbf{x}_t, \quad \textit{started at } y_0^0 = y_0, \tag{11}$$

$$dy_t^n = f(y_t^n) \diamond d\mathbf{x}_t, \quad \textit{started at } y_{\tau_n}^n = \mathcal{T}\left(y_{\tau_n-}^{n-1}\right). \tag{12}$$

Existence and uniqueness of solutions to Event RDEs is proven in the same way as for Event SDEs. Indeed, under the usual assumption that the vector fields $f$ are $\mathrm{Lip}^\gamma$, for $\gamma > p$, a unique solution to (11) exists. In fact, the solution map $y_s \times (s, t) \mapsto y_t$ is a diffeomorphism for every fixed $0 \leq s < t \leq T$ (see, e.g., Theorem 3.13 in [7]). It follows that we can iteratively define a unique sequence of solutions $y^n \in D_p([t_n, T], \mathbb{R}^d)$. Finally, as mentioned in Remark A.2, if the driving rough path $\mathbf{x}$ is the Marcus lift of a semi-martingale, the inter-event solutions agree almost surely with the solutions to the corresponding Marcus SDE.

**Theorem A.2.** *Under Assumptions 3.1-3.2, there exists a unique solution $(y, (\tau_n)_{n=1}^N)$ to the Event RDE of Definition A.9. Furthermore, if $\mathbf{x}$ is the Marcus lift of a Brownian motion, the solution coincides almost surely with the solution to the corresponding Event SDE as given in Def. 3.1.*

Hence, the Event SDEs considered in the main text are special cases of Event RDEs driven by the Marcus lift of a Brownian motion. Yet, the more general formulation of Event RDEs allows to treat, using the same mathematical machinery of rough path theory a much larger family of driving noises such as fractional Brownian motion or even smooth controls. Also, since the driving rough path is allowed to be càdlàg, the model class given by Def. A.9 includes cases where the inter-event dynamics are given by Marcus SDEs driven by general semi-martingales.

## B  Proof of Theorem 3.2

The proof of Theorem 3.2 presented below covers the case where $(y, (\tau_n)_{n=1}^N)$ is the solution to an Event RDE. Throughout we consider vector fields $\mu \in \mathrm{Lip}^1, \sigma \in \mathrm{Lip}^{2+\epsilon}$ and specialise to Event RDEs where the inter-event dynamics are given by

$$dy_t^n = \mu(y_t^n)dt + \sigma(y_t^n) \diamond d\mathbf{x}_t, \tag{13}$$

where $\mathbf{x} \in \Omega_p^D(\mathbb{R}^d)$. The notation above deserves some clarification. One can define the vector field $f = (\mu, \sigma)$ and the Young pairing $\tilde{\mathbf{x}}_t$ of $\mathbf{x}$ and $h_t = t$. Assuming $\mu \in \mathrm{Lip}^{2+\epsilon}$ we can then view $y_t^n$ as the unique solution to the Marcus RDE

$$dy_t^n = f(y_t^n) \diamond \tilde{\mathbf{x}}_t.$$

Alternatively, if one is not ready to impose the added regularity on the drift $\mu$, one can view 13 as a RDE with drift as in Ch. 12 in [17]. To accommodate this more general case where the path driving the diffusion term might be 1) càdlàg and 2) is not restricted to be the rough path lift of a semi-martingale, we shall need the following two additional assumptions:

*Assumption* B.1. For any $n \in [N]$, there exists a non-empty interval $I_n = (\tau_n - \delta_n, \tau_n + \delta_n)$ such that $\mathbf{x}$ is continuous over $I_n$. In other words, the càdlàg rough path $\mathbf{x}$, does not jump in small intervals around the event times $(\tau_n)$.

*Assumption* B.2. For all $0 \le n \le N$ we define $s_n = \tau_n - \delta_n/2$ and $t_n = \tau_{n+1} + \delta_{n+1}/2$. It holds that $\mathbf{x} \in \Omega_{0,p}^D([s_n, t_n], \mathbb{R}^d)$, i.e., $\mathbf{x}$ is a geometric $p$-rough path on the intervals $[s_n, t_n]$.

*Remark* B.1. Note that Assumption B.1 trivially holds if $\mathbf{x}$ is continuous. Otherwise, it is enough to assume, e.g., that $\mathbf{x}$ is the Marcus lift of a *finite activity* Lèvy process. Furthermore, by the properties of the metric $\alpha_p$, if $\mathbf{x}$ is the canonical Marcus lift of a semi-martingale $x \in D_p([s, t], \mathbb{R}^{d-1})$, then there exists a sequence $(x^m)$ of piece-wise linear paths $x^m \in C_0^1([0, T], \mathbb{R}^{d-1})$ such that

$$\alpha_{p,[s_n,t_n]}(x^m, \mathbf{x}) \to 0 \quad \text{as } m \to \infty \quad \text{a.s.}$$

See, e.g. [7, Example 4.21]. The setting of Section 3.3 is therefore a special case of the setting considered here and Theorem 3.2 follows from the proof below.

We shall need two technical lemmas for the proof of 3.2

**Lemma B.1.** *Assume that Assumptions 3.1-3.5 and B.1-B.2 are satisfied. Then, there exists an open ball $B_0 \subset O$ such that the following holds:*

1. *For all $a \in B_0$, $|\tau(a)| = N$.*

2. *For any $n \in [N]$, the maps*

$$B_0 \ni a \mapsto \left(\tau_n(a), y_{\tau_n(a)}^{n-1}(a)\right) \quad \text{are continuous.}$$

3. *For the sequence $(x^m)$ as given in Assumption B.2 and $(y^m, (\tau_n^m)_{n=1}^N)$ the corresponding Event RDE solution, for all $n \in [N]$, it holds that*

$$\lim_{m \to \infty} \sup_{a \in B_0} \left(|\tau_n^m(a) - \tau_n(a)| + \left|y_{\tau_n^m(a)}^{m,n-1}(a) - y_{\tau_n(a)}^{n-1}(a)\right|\right) = 0.$$

*Proof.* Recall that $\Phi(y, s, t; \mathbf{x})$ is the solution map or flow of the differential equation

$$dy_u = f(y_u) \diamond d\tilde{\mathbf{x}}_u, \quad y_s = y$$

evaluated at time $t$. The first step will be to prove continuity at $y_0$. In particular, let $y_0^m \in O$ approach $y_0$ for $m$ going to infinity and denote the solutions to the corresponding Event RDEs by $\left(y^m, (\tau_n^m)_{n=1}^{N_m}\right)$. We claim that $\lim_{m \to \infty} N_m = N$ and

$$\lim_{m \to \infty} \tau_n^m = \tau_n, \quad \lim_{m \to \infty} y_{\tau_n^m}^{m,n-1} = y_{\tau_n}^n.$$

To see this, note that, by Theorem A.1, there exists a sequence $\lambda_m \in \Lambda$ of continuous reparameterisations such that $|\lambda_m| \to 0$ and

$$\sup_{(s,t) \in \Delta_T} |\Phi(y_0, s, t; \mathbf{x}) - \Phi(y_0^m, \lambda^m(s), \lambda^m(t); \mathbf{x})| \to 0 \tag{14}$$

for $m \to \infty$. Note, furthermore, that $\Phi(y_0^m, s, t; \mathbf{x} \circ \lambda_m) = \Phi(y_0^m, \lambda_m(s), \lambda_m(t); \mathbf{x})$ for all $(s, t) \in \Delta_T$. We let $\left(\tilde{y}^m, (\tilde{\tau}_n^m)_{n=1}^{N_m}\right)$ be the solution to the Event RDE where $(y_0, \mathbf{x})$ is replaced by $(y_0^m, \mathbf{x} \circ \lambda_m)$. It suffices to prove that, for all $1 \le n \le N$,

$$\lim_{m \to \infty} \tilde{\tau}_n^m = \tau_n, \quad \lim_{m \to \infty} \tilde{y}_{\tilde{\tau}_n^m}^{m,n-1} = y_{\tau_n}^n. \tag{15}$$

Indeed, since $\tilde{\tau}_n^m = \lambda_m^{-1}(\tau_n^m)$ and $|\lambda_m| \to 0$, it then follows that $\tau_n^m \to \tau_n$ for $m \to \infty$. Furthermore, we have $\tilde{y}_{\tilde{\tau}_n^m}^{m,n-1} = y_{\tau_n^m}^{m,n-1}$.

We shall proof (15) using an inductive argument. We have that

$$y_t^0 = \Phi(y_0, 0, t; \mathbf{x}), \quad \forall t \in [0, \tau_1],$$

$$\tilde{y}_t^{m,0} = \Phi(y_0^m, 0, t; \mathbf{x} \circ \lambda_m), \quad \forall t \in [0, \tilde{\tau}_1^m].$$

Now fix some $0 < \epsilon < \delta_1$ where $\delta_1$ is given in Assumption B.1. Note that $|\mathcal{E}(y_t^0)| > 0$ for all $t \in [0, \tau_1 - \epsilon]$ and therefore, by (14), it follows that there exists an $m_0 \in \mathbb{N}$ such that, for all $m \geq m_0$,

$$\inf_{t \in [0, \tau_1 - \epsilon]} |\mathcal{E}(\Phi(y_0^m, 0, t; \mathbf{x} \circ \lambda_m))| > 0$$

so that $\tilde{\tau}_1^m \geq \tau_1 - \epsilon$. Next, for some small $0 < \eta < \epsilon$, Assumption 3.4 and the Mean Value Theorem imply the existence of $a_\eta^+ = r_\eta^+ y_{\tau_1}^0 - (1 - r_\eta^+) y_{\tau_1 + \eta}^0$, and $a_\eta^- = r_\eta^- y_{\tau_1 - \eta}^0 + (1 - r_\eta^-) y_{\tau_1}^0$ with $r_\eta^+, r_\eta^- \in (0, 1)$ such that

$$\mathcal{E}\left(y_{\tau_1 + \eta}^0\right) = \mathcal{E}\left(y_{\tau_1}^0\right) + \nabla \mathcal{E}(a_\eta^+) \int_{\tau_1}^{\tau_1 + \eta} \mu(y_s^0) dy_s,$$

$$\mathcal{E}\left(y_{\tau_1 - \eta}^0\right) = \mathcal{E}\left(y_{\tau_1}^0\right) - \nabla \mathcal{E}(a_\eta^-) \int_{\tau_1 - \eta}^{\tau_1} \mu(y_s^0) dy_s,$$

But then, by Assumption 3.5 and the fact that $\mathcal{E}(y_{\tau_1}^0) = 0$, for $\eta$ small enough, $\mathcal{E}(y_{\tau_1 + \eta}^0)$ and $\mathcal{E}(y_{\tau_1 - \eta}^0)$ must lie on different sides of 0. Assumption B.1 and eq. (14) then yield the existence of a $m_1 \geq m_0$ such that $\tilde{\tau}_1^m \leq \tau_1 + \eta \leq \tau_1 + \epsilon$ and $\inf_{t \in [0, \tau_1 + \eta]} |\mathcal{E}(\tilde{y}_t^{m,0})| > 0$ for all $m \geq m_1$. It follows that $\tilde{\tau}_1^m \to \tau_1$. Finally, note that

$$\left| \tilde{y}_{\tilde{\tau}_1^m}^{m,0} - y_{\tau_1}^0 \right| \leq \left| \tilde{y}_{\tilde{\tau}_1^m}^{m,0} - y_{\tilde{\tau}_1^m}^0 \right| + \left| y_{\tilde{\tau}_1^m}^0 - y_{\tau_1}^0 \right|.$$

Another application of (14) shows that the first term on the right hand side goes to 0 for $m \to \infty$ and second term vanishes by Assumption B.1.

To prove the inductive step, assume that (15) holds for $i \leq n$. For all $t \in [\tau_n, \tau_{n+1}]$ it holds that

$$\tilde{y}_t^{m,n} = \Phi\left(\mathcal{T}\left(\tilde{y}_{\tilde{\tau}_n^m}^{m,n-1}\right), \tilde{\tau}_n^m, t; \mathbf{x} \circ \lambda_m\right), \quad y_t^n = \Phi\left(\mathcal{T}\left(y_{\tau_n}^{n-1}\right), \tau_n, t; \mathbf{x}\right)$$

and, since $\tilde{y}_{\tilde{\tau}_n^m}^{m,n-1} \to y_{\tau_n}^{n-1}$, $\tilde{\tau}_n^m \to \tau_n$, and $\mathcal{T}$ is continuous,

$$\lim_{m \to \infty} \sup_{t \in [\tau_n, T]} \left| \Phi\left(\mathcal{T}\left(\tilde{y}_{\tilde{\tau}_n^m}^{m,n-1}\right), \tilde{\tau}_n^m, t; \mathbf{x} \circ \lambda_m\right) - \Phi\left(\mathcal{T}\left(y_{\tau_n}^{n-1}\right), \tau_n, t; \mathbf{x}\right) \right| = 0$$

whence the same argument as above proves that (15) also holds for $n + 1$. This completes the proof of the claim.

Now, by continuity at $y_0$, it follows that there exists some small $r > 0$ such that for all $a \in B_r(y_0)$ it holds that $|\tau(a)| = N$ and $\tau_n(a) \in (\tau_n - \delta_n/2, \tau_n + \delta_n/2)$ for all $n \in [N]$ where $\delta_n$ is as in Assumption B.1. Furthermore, since Assumption 3.1-3.5 and B.1-B.2 still hold for $a \in B_r(y_0)$, the same argument as above can be applied to show that $\tau_n(a)$ and $y_{\tau_n(a)}^{n-1}(a)$ are continuous at $a$. This proves parts 1 and 2.

To prove part 3 we employ a similar induction argument to the one above. First, note that, by Theorem A.1, there exists a constant $C > 0$ not depending on $x$ such that

$$\alpha_{p, [0, t_0]}\left(y^{m,0}(a), y^0(a)\right) \leq C \alpha_{p, [0, t_0]}\left(x^m, \mathbf{x}\right).$$

Since the latter term does not depend on $y$ and goes to 0 for $m$ going to infinity, we find that

$$\lim_{m \to \infty} \sup_{a \in B_r(y_0)} \alpha_{p, [0, t_1]}\left(y^{m,0}(a), y^0(a)\right) = 0. \tag{16}$$

Recall, $y^{\delta,0}(a)$ is the continuous path obtained by the Marcus interpolation with $\delta r_k$ instead of $r_k$ and similarly for $y^{m,\delta,0}(a)$. Note that $y^{m,\delta,0}(a) = y^{m,0}(a)$ by continuity. Letting $\tau_1^m(a)$ and $\tau_1^\delta(a)$ denote the first event time of $y^{m,0}(a)$ and $y^{\delta,0}(a)$ respectively, we have, for all $m \in \mathbb{N}$

$$\sup_{a \in B_r(x_0)} |\tau_1^m(a) - \tau_1(a)| \leq \sup_{a \in B_r(y_0)} \lim_{\delta \to 0} \left( \left| \tau_1^m(a) - \tau_1^\delta(a) \right| + \left| \tau_1^\delta(a) - \tau_1(a) \right| \right).$$

Now, let $B_0 = B_r(y_0)$. Since $\tau_1(a) \in (\tau_1 - \delta_1/2, \tau_1 + \delta_1/2)$ for all $a \in B_0$ and $\mathbf{x}$ is continuous over this interval, it follows that $\left|\tau_1^\delta(a) - \tau_1(a)\right|$ goes to 0 as $\delta \to 0$ for each $a \in B_0$. Furthermore, by definition of the metric $\alpha_p$, eq. (16), and the fact that $y_0^{m,0}(a) = a = y_0^0(a)$, for each $a \in B_0$, a similar argument as the one employed in the beginning of the proof then shows that $\left|\tau_1^m(a) - \tau_1^\delta(a)\right| \to 0$ as $\delta \to 0$ and, thus, $\lim_{m \to \infty} \sup_{a \in B_0} \left|\tau_1^m(a) - \tau_1(a)\right| = 0$. Finally, starting from the inequality

$$\left|y_{\tau_1^m(a)}^{m,0}(a) - y_{\tau_1(a)}^0(a)\right| \le \left|y_{\tau_1^m(a)}^{m,0}(x) - y_{\tau_1^\delta(a)}^{\delta,0}(a)\right| + \left|y_{\tau_1^\delta(a)}^{\delta,0}(a) - y_{\tau_1(a)}^0(a)\right|$$

and taking the limit as $\delta \to 0$ and then the supremum over $x \in B_0$ on both sides, we can argue in exactly the same way to show that part 3 holds for $n = 1$. We can then argue by induction, just as in the first part of the proof, to show that it holds for all subsequent event times as well. Thus, the set $B_0$ satisfies all the stated requirements. $\qquad\square$

**Lemma B.2.** *Let Assumption B.1 hold and $x^m$ be as in Assumption B.2. Then, for all $n \in [N]$ and $p' > p$,*

$$\lim_{m \to \infty} d_{p',[s,t]}\left(x^m, \mathbf{x}\right) = 0, \quad \text{for any } \tau_n - \delta_n/2 \le s < t \le \tau_n + \delta_n/2.$$

*Proof.* Fix some $n \in [N]$, $p' > p$ and $\tau_n - \delta_n/2 \le s < t \le \tau_n + \delta_n/2$. Note that, for any continuous reparameterisation $\lambda \in \Lambda$, $m \in \mathbb{N}$, and $\delta > 0$, it holds that

$$d_{p',[s,t]}(x^m, \mathbf{x}) \le d_{p',[s,t]}(x^m, x^m \circ \lambda) + d_{p',[s_n,t_n]}(x^m \circ \lambda, \hat{\mathbf{x}}^\delta) + d_{p',[s,t]}(\hat{\mathbf{x}}^\delta, \mathbf{x}),$$

where $\hat{\mathbf{x}}^\delta$ is the Marcus interpolation of $\mathbf{x}$ over the interval $[s_n, t_n]$. Taking the infimum over $\lambda \in \Lambda$ and the limit as $\delta \to 0$ on both sides, we obtain

$$d_{p',[s,t]}(x^m, \mathbf{x}) \le \alpha_{p',[s_n,t_n]}(x^m, \mathbf{x}) + \lim_{\delta \to 0} d_{p',[s,t]}(\hat{\mathbf{x}}^\delta, \mathbf{x}).$$

The first term on the right hand side goes to 0 as $m \to \infty$ by Assumption B.2. Furthermore, since, by Assumption B.1, $\mathbf{x}$ is continuous on $(\tau_n - \delta_n, \tau_n + \delta_n)$, it follows that $d_{\infty,[s,t]}(\hat{\mathbf{x}}^\delta, \mathbf{x})$ goes to 0 for $\delta \to \infty$. But the result then follows from Proposition 8.15 and Lemma 8.16 in [17]. $\qquad\square$

*Proof of Theorem 3.2. Step 1:* Assume that $x \in C_1([0,T], \mathbb{R}^{d-1})$. By [17, Theorem 4.4], the Jacobian $\partial y_t^0$ exists and satisfies (7) for all $t \in [0, \tau_1)$. We shall prove that relations (6) and (7) hold for all $n \in [N]$ by induction. Thus, assume that $\partial y_t^k$ and $\partial \tau_k$ exist for all $t \in [\tau_k, \tau_{k+1})$ and $k \le n - 1$ and satisfy the stated relations. To emphasise the dependence on the initial condition, we will sometimes use the notation $y^n = y^n(y_0)$ and $\tau_n = \tau_n(y_0)$ for the solution of the Event RDE started at $y_0$. We want to show that, for arbitrary $h \in \mathbb{R}^e$, the following limits

$$\lim_{\epsilon \to 0} \frac{\tau_n^\epsilon - \tau_n}{\epsilon} \quad \text{and} \quad \lim_{\epsilon \to 0} \frac{y_t^{n,\epsilon} - y_t^n}{\epsilon} \quad \text{for } t \in [\tau_n, \tau_{n+1})$$

exist and satisfy the stated expressions, where $\tau_n^\epsilon = \tau_n(y_0 + h\epsilon)$ and $y^{n,\epsilon} = y^n(y_0 + h\epsilon)$.

For any $\epsilon > 0$, because $\mathcal{E}$ is continuously differentiable, the Mean Value Theorem implies that there exists $c_\epsilon \in \mathbb{R}^e$ on the line connecting $y_{\tau_n}^{n-1}$ to $y_{\tau_n^\epsilon}^{n-1}$ and another $c_\epsilon' \in \mathbb{R}^e$ on the line connecting $y_{\tau_n^\epsilon}^{n-1,\epsilon}$ to $y_{\tau_n^\epsilon}^{n-1}$ such that

$$\mathcal{E}\left(y_{\tau_n}^{n-1}\right) = \mathcal{E}\left(y_{\tau_n^\epsilon}^{n-1}\right) + \nabla\mathcal{E}(c_\epsilon)\left(y_{\tau_n}^{n-1} - y_{\tau_n^\epsilon}^{n-1}\right)$$

$$= \mathcal{E}\left(y_{\tau_n^\epsilon}^{n-1}\right) + \nabla\mathcal{E}\left(c_\epsilon\right)\left(\mu(y_{\tau_n}^{n-1})(\tau_n - \tau_n^\epsilon) + \sigma\left(y_{\tau_n}^{n-1}\right)(x_{\tau_n} - x_{\tau_n^\epsilon}) + o(|\tau_n - \tau_n^\epsilon|)\right),$$

$$\mathcal{E}\left(y_{\tau_n^\epsilon}^{n-1,\epsilon}\right) = \mathcal{E}\left(y_{\tau_n^\epsilon}^{n-1}\right) + \nabla\mathcal{E}(c_\epsilon')\left(y_{\tau_n^\epsilon}^{n-1,\epsilon} - y_{\tau_n^\epsilon}^{n-1}\right)$$

$$= \mathcal{E}\left(y_{\tau_n^\epsilon}^{n-1}\right) + \nabla\mathcal{E}(c_\epsilon')\left(\epsilon\left(\partial y_{\tau_n^\epsilon}^{n-1}\right)h + o(\epsilon)\right),$$

where the last equality follows from the induction hypothesis. We have $\mathcal{E}(y_{\tau_n}^{n-1}) = 0 = \mathcal{E}(y_{\tau_n^\epsilon}^{n-1,\epsilon})$. Thus, by rearranging, we find that

$$\frac{\tau_n^\epsilon - \tau_n}{\epsilon} = -\frac{\nabla\mathcal{E}(y_{\tau_n}^{n-1})\partial y_{\tau_n}^{n-1}h}{\nabla\mathcal{E}(y_{\tau_n}^{n-1})\left(\mu(y_{\tau_n}^{n-1}) + \sigma(y_{\tau_n}^{n-1})\frac{x_{\tau_n} - x_{\tau_n^\epsilon}}{\tau_n - \tau_n^\epsilon}\right)} + o(1)$$

$$= -\frac{\nabla\mathcal{E}(y_{\tau_n}^{n-1})\partial y_{\tau_n}^{n-1}h}{\nabla\mathcal{E}(y_{\tau_n}^{n-1})\mu(y_{\tau_n}^{n-1})} + o(1)$$

where the second equality follows from Assumptions 3.4 and 3.5.

Assume for now that $\tau_n^\epsilon < \tau_n$. By another application of the Mean Value Theorem, there exists $c_\epsilon \in \mathbb{R}^e$ on the line connecting $y_{\tau_n}^{n-1}$ to $y_{\tau_n^\epsilon}^{n-1}$ such that

$$
\begin{aligned}
y_{\tau_n}^{n,\epsilon} - y_{\tau_n}^n &= y_{\tau_n}^{n,\epsilon} - \mathcal{T}\left(y_{\tau_n}^{n-1}\right)\\
&= y_{\tau_n}^{n,\epsilon} - \mathcal{T}\left(y_{\tau_n^\epsilon}^{n-1}\right) - \nabla\mathcal{T}(c_\epsilon)(y_{\tau_n}^{n-1} - y_{\tau_n^\epsilon}^{n-1})\\
&= y_{\tau_n^\epsilon}^{n,\epsilon} + \mu(y_{\tau_n}^{n,\epsilon})(\tau_n - \tau_n^\epsilon) + \sigma\left(y_{\tau_n}^{n,\epsilon}\right)(x_{\tau_n} - x_{\tau_n^\epsilon}) - \mathcal{T}\left(y_{\tau_n^\epsilon}^{n-1}\right)\\
&\quad - \nabla\mathcal{T}(c_\epsilon)\left(\mu(y_{\tau_n}^{n-1})(\tau_n - \tau_n^\epsilon) + \sigma\left(y_{\tau_n}^{n-1}\right)(x_{\tau_n} - x_{\tau_n^\epsilon}) + o(|\tau_n - \tau_n^\epsilon|)\right)\\
&= \mathcal{T}\left(y_{\tau_n^\epsilon}^{n-1,\epsilon}\right) - \mathcal{T}\left(y_{\tau_n^\epsilon}^{n-1}\right) + \left(\mu(y_{\tau_n}^{n,\epsilon}) - \nabla\mathcal{T}(c_\epsilon)\mu(y_{\tau_n}^{n-1})\right)(\tau_n - \tau_n^\epsilon)\\
&\quad + \left(\sigma(y_{\tau_n}^n) - \nabla\mathcal{T}(c_\epsilon)\sigma(y_{\tau_n}^{n-1})\right)(x_{\tau_n} - x_{\tau_n^\epsilon}) + o(|\tau_n - \tau_n^\epsilon|)
\end{aligned}
$$

Therefore

$$
\frac{y_{\tau_n}^{n,\epsilon} - y_{\tau_n}^n}{\epsilon} = \nabla\mathcal{T}(y_{\tau_n}^{n-1})\partial y_{\tau_n}^{n-1}h + \left(\mu(y_{\tau_n}^n) - \nabla\mathcal{T}(y_{\tau_n}^{n-1})\mu(y_{\tau_n}^{n-1})\right)\partial\tau_n h + o(1)
$$

where we used Assumption 3.3, the chain rule and the existence of $\partial\tau_n$. Finally, for any $t \in (\tau_n, \tau_{n+1}]$, equation (7) follows from the fact that we can write $y_t^n = \Phi(y_s^n, s, t, x)$ for all $\tau_n \leq s < t$. In particular, by the chain rule, we find that

$$
\partial y_t^n = \left[\partial_{y_s^n}\Phi(y_s^n, s, t)\partial y_s^n\right]_{s=\tau_n} = \partial_{y_{\tau_n}^n} y_t^n \left[\partial y_s^n\right]_{s=\tau_n}.
$$

*Step 2:* Consider now the general case of $\mathbf{x} \in \Omega_p(\mathbb{R}^e)$ and let $(y^m, (\tau_{n_m}^m)_{n_m}^{N_m})$ denote the solution to the Event RDE where $\mathbf{x}$ is replaced by the piece-wise linear approximation $x^m$. With $\partial y_t^{n,m}$ and $\partial\tau_n^m$ denoting the corresponding derivatives, we saw in the previous step that both exist and satisfy (6)-(7). We let $R_t^n$ and $\rho_n$ denote the right hand side of (7) and (6) respectively. This step consists of proving that, for $n \in [N]$ and $t \in (\tau_n, t_n)$,

$$
\lim_{m\to\infty}\left\{|\tau_n^m - \tau_n| + |y_t^{m,n} - y_t^n|\right\} = 0 \tag{17}
$$

and for some open ball $B_0$ around $y_0$

$$
\lim_{m\to\infty}\sup_{a\in B_0}\left\{|\partial\tau_n^m(a) - \rho_n(a)| + \|\partial y_t^{m,n}(a) - R_t^n(a)\|\right\} = 0. \tag{18}
$$

By Lemma B.1 and continuity of $\mathcal{T}$ we have that $\mathcal{T}(y_{\tau_n^m}^{m,n-1})$ converges to $\mathcal{T}(y_{\tau_n}^{n-1})$ and $\tau_n^m$ converges to $\tau_n$ as $m \to +\infty$. Then, because $y_t^n = \Phi(\mathcal{T}(y_{\tau_n}^{n-1}), \tau_n, t; \mathbf{x})$ and $y_t^{m,n} = \Phi(\mathcal{T}(y_{\tau_n^m}^{m,n-1}), \tau_n^m, t; x^m)$, equation (17) follows from, Lemma B.2 and Corollary 11.16 in [17]. In fact, since $B_0$ was constructed in Lemma B.1 in such a way that $\tau_n(a) < t_n$ for all $a \in B_0$ we also get that

$$
\lim_{m\to\infty}\sup_{a\in B_0}\left\|\partial_{y_{\tau_n(a)}^n(a)}\Phi\left(y_{\tau_n(a)}^n(a), \tau_n(a), t; \mathbf{x}\right) - \partial_{y_{\tau_n^m(a)}^{m,n}(a)}\Phi\left(y_{\tau_n^m(a)}^{m,n}(a), \tau_n^m(a), t; x^m\right)\right\| = 0.
$$

by the same corollary in [17]. Thus, to prove (18), it suffices to show that, for all $n \in \{1, ..., N\}$,

$$
\lim_{m\to\infty}\sup_{a\in B_0}\left\|\partial y_{\tau_n^m(a)}^{m,n-1}(a) - R_{\tau_n(a)}^{n-1}(a)\right\| = 0.
$$

We shall prove it using another inductive argument starting with $n = 1$. In this case it suffices to show that

$$
\lim_{m\to\infty}\sup_{a\in B_0}\|\partial_a\Phi(a, 0, \tau_1(a); \mathbf{x}) - \partial_a\Phi(a, 0, \tau_1^m(a); x^m)\| = 0.
$$

By [7, Theorem 3.3] we know that the above holds if $\tau_1^m(a)$ and $\tau_1(a)$ are replaced by $\tau_1 + \delta_1/2$. Now let $\Phi^{-1}$ be the reverse of the flow map $\Phi$, that is,

$$
\Phi^{-1}(a_1, s, t; \mathbf{x}) = a_0 \Leftrightarrow \Phi(a_0, s, t; \mathbf{x}) = a_1.
$$

From Lemma B.1 it follows that $y^0_{\tau_1(a)}(a) = \Phi^{-1}(y^0_{t_0}(a), \tau_1(a), t_0; \mathbf{x})$ and, for $m$ large enough, $y^{m,0}_{\tau_1^m(a)}(a) = \Phi^{-1}(y^{m,0}_{t_0}(a), \tau_1^m(a), t_0; x^m)$. But the result then follows from Lemma B.2 and [17, Corollary 11.16]. To prove the inductive step, assume that (18) holds for all $i \leq n-1$. Again, by inspecting (6) and (7) and using the inductive assumption, one finds that it is enough to show that

$$\lim_{m \to \infty} \sup_{a \in B_0} \left\| \partial_{y^{n-1}_{\tau_{n-1}}} \Phi\left(y^{n-1}_{\tau_{n-1}}, \tau_{n-1}, \tau_n; \mathbf{x}\right) - \partial_{y^{m,n-1}_{\tau_{n-1}^m}} \Phi\left(y^{m,n-1}_{\tau_{n-1}^m}, \tau_{n-1}^m, \tau_n^m; x^m\right) \right\| = 0,$$

where we suppressed the dependence on $a$ for notational simplicity. This is done exactly as for $y^0$ and completes the proof of Step 2.

*Step 3:* The third and final step is to combine Step 1 and 2 to finish the proof. So far we have proven that 1) the theorem holds for continuous paths of bounded variation and 2) $(\tau_n^m, y_t^{m,n})$ converges to $(\tau_n, y_t^n)$ and $(\partial \tau_n^m(a), \partial y_t^{m,n}(a))$ converges uniformly to $(\rho_n(a), R_t^n(a))$ over $a \in B_0$ for all $t \in (\tau_n, t_n)$ and $n \in [N]$. From these results it immediately follows that $(\tau_n(a), y_t^n(a))$ is differentiable at $a = y_0$ with derivatives given by $(\rho_n(y_0), R_t(y_0))$ for all $t \in (\tau_n, t_n)$. What is left to show then, is that this also holds for all other $t$. But this follows immediately from the chain rule upon realizing that, for any $\tau_n < s < \tau_n + \delta_n/2 < t < \tau_{n+1}$,

$$y_t^n = \Phi(y_s^n, s, t; \mathbf{x}) \Rightarrow \partial y_t^n = \partial_{y_s^n} \Phi(y_s^n, s, t; \mathbf{x}) R_s^n(y_0) = R_t^n(y_0).$$

$\square$

## C Kernel methods

We give here a brief outline of some of the most central concepts related to kernel methods. For a more in-depth introduction we refer the reader to [45, 55, 3]. Let $\mathcal{X}$ be a topological space. We shall in this paper only be concerned with positive definite kernels, that is, symmetric functions $k : \mathcal{X} \times \mathcal{X} \to \mathbb{R}$ for which the Gram matrix is positive definite. To such a kernel one may associate a feature map $\mathcal{X} \to \mathbb{R}^{\mathcal{X}}$ such that $x \mapsto k_x = k(x, \cdot)$. A reproducing kernel Hilbert space (RKHS) is a Hilbert space $\mathcal{H} \subset \mathbb{R}^{\mathcal{X}}$ such that the evaluation functionals, $ev_x : f \mapsto f(x)$, are bounded for each $x \in \mathcal{X}$. For all positive definite kernels there is a unique RKHS $\mathcal{H} \subset \mathbb{R}^{\mathcal{X}}$ such that $f(x) = \langle k_x, f \rangle_{\mathcal{H}}$ for all $f \in \mathcal{H}$ and $x \in \mathcal{X}$. This is also known as the *reproducing property*. Furthermore, with $H$ denoting the linear span of $\{k_x \mid x \in \mathcal{X}\}$, it holds that $\bar{H} = \mathcal{H}$, i.e., $H$ is dense in $\mathcal{H}$. Two important properties of kernels are *characteristicness* and *universality*.

**Definition C.1.** *Let $k : \mathcal{X} \times \mathcal{X} \to \mathbb{R}$ be a positive definite kernel. Denote by $H$ the linear span of $\{k_x \mid x \in \mathcal{X}\}$ and let $\mathcal{F} \subset \mathbb{R}^{\mathcal{X}}$ be a topological vector space containing $H$ and such that the inclusion map $\iota : H \to \mathcal{F}$ is continuous.*

- *We say that $k$ is universal to $\mathcal{F}$ if the embedding of $\iota : H \to \mathcal{F}$ is dense.*

- *We say that $k$ is characteristic to $\mathcal{F}'$ if the embedding $\mu : \mathcal{F}' \to H', D \mapsto D|_H$ is injective*

*Remark* C.1. This definition is the one used in [8] and is more general then the one usually encountered. Note that in many cases (all the cases considered here, in fact) $\mathcal{F}'$ will contain the set of probability measures on $\mathcal{X}$ in which case $k$ being characteristic implies that the *kernel mean embedding* $\mu \mapsto \mathbb{E}_{X \sim \mu} k_X(\cdot)$ is injective.

*Remark* C.2. Often times, instead of starting with the kernel function $k$ and then obtaining the RKHS, one starts with a feature map $F : \mathcal{X} \to \mathcal{H}$ into a RKHS and then defines the kernel as the inner product in that Hilbert space, i.e., $k(x, y) = \langle F(x), F(y) \rangle_{\mathcal{H}}$. In such cases, it makes sense to ask whether there are equivalent notions of $F$ being universal and characteristic. This is indeed the case and the definition is almost the same as above. We refer to Definition 6 in [8] for a precise statement.

### C.1 Marcus signature kernel

The definition of the signature kernel requires an initial algebraic setup. Let $\langle \cdot, \cdot \rangle_1$ be the Euclidean inner product on $\mathbb{R}^d$. Denote by $\otimes$ the standard outer product of vector spaces. For any $n \in \mathbb{N}$, we denote by $\langle \cdot, \cdot \rangle_n$ on $(\mathbb{R}^d)^{\otimes n}$ the canonical Hilbert-Schmidt inner product defined for any $\mathbf{a} = (a_1, \ldots, a_n)$ and $\mathbf{b} = (b_1, \ldots, b_n)$ in $(\mathbb{R}^d)^{\otimes n}$ as $\langle \mathbf{a}, \mathbf{b} \rangle_n = \prod_{i=1}^n \langle a_i, b_i \rangle_1$. The inner product $\langle \cdot, \cdot \rangle_n$

on $(\mathbb{R}^d)^{\otimes n}$ can then be extended by linearity to an inner product $\langle \cdot, \cdot \rangle$ on $\tilde{T}((\mathbb{R}^d))$ defined for any $\mathbf{a} = (1, a_1, \dots)$ and $\mathbf{b} = (1, b_1, \dots)$ in $\tilde{T}((\mathbb{R}^d))$ as $\langle \mathbf{a}, \mathbf{b} \rangle = 1 + \sum_{n=1}^{\infty} \langle a_n, b_n \rangle_n$.

To begin with, let $\mathcal{X} = D_1([0, T], \mathbb{R}^d)$. If $x \in \mathcal{X}$ is càdlàg path, we can define the *Marcus signature* in the spirit of Marcus SDEs [42, 43] as the signature of the *Marcus interpolation* of $x$. This interpolation, denoted by $\hat{x}$, is the continuous path on $[0, T]$ obtained from $x$ by linearly traversing the jumps of $x$ over added fictitious time $r > 0$ and then reparameterising so that the path runs over $[0, T]$ instead of $[0, T + r]$. The general construction is given in Appendix A. If $x$ is continuous, $x$ and $\hat{x}$ coincide; thus, without any ambiguity, we can define the Marcus signature $S(x)$ of a general bounded variation càdlàg path as the tensor series described above, but replacing $x$ with $\hat{x}$ (see also the definition in A.2).

Since the signature is invariant to certain reparameterisations (Proposition A.1), it is not an injective map. Injectivity is a crucial property required to ensure characteristicness of the resulting signature kernel that we will introduce next. One way of overcome this issue is to to augment a path $x$ with a time coordinate resulting in the path $\tilde{x} = (x, t)^5$. The Marcus signature kernel is then naturally defined as the map $k : \mathcal{X} \times \mathcal{X} \to \mathbb{R}$ such that $k(x, y) = \langle S(\tilde{x}), S(\tilde{y}) \rangle$ for any $x, y \in \mathcal{X}$. As stated in Theorem C.1, this kernel is universal on compact subsets $K \subset \mathcal{X}$ and, equivalently, characteristic to the space of regular Borel measures on $K$. However, these properties do not generalize to the whole space $C_b(\mathcal{X}, \mathbb{R})$ of bounded continuous functions from $\mathcal{X}$ to $\mathbb{R}$.

In [8] the authors address this issue in the case of continuous paths by introducing the so-called *robust signature*. They define a *tensor normalization* as a continuous injective map

$$\Lambda : \tilde{T}\left((\mathbb{R}^d)\right) \to \left\{ \mathbf{a} \in \tilde{T}\left((\mathbb{R}^d)\right) \mid \|\mathbf{a}\| \leq R \right\}$$

for some $R > 0$ and such that $\Lambda(\mathbf{a}) = (\mathbf{a}^0, \lambda(\mathbf{a}) a_1, \lambda(\mathbf{a})^2 a_2, \dots)$ for some $\lambda : \tilde{T}\left((\mathbb{R}^d)\right) \to (0, \infty)$.

Now, let $p \in [1, 3)$ and take $C_0^1(\mathbb{R}^d)$ to be the space of absolutely continuous functions on $\mathbb{R}^d$. Recall that $\Omega_{0,p}^D(\mathbb{R}^d)$ is the closure of $C_0^1(\mathbb{R}^d)$ in $\Omega_p^D(\mathbb{R}^d)$ under the metric $\alpha_p$. Throughout we let $\mathcal{X} = \Omega_{0,p}^D(\mathbb{R}^d)$ be a metric space equipped with $\alpha_p$. Naturally, we can then define the signature kernel on $\mathcal{X}$ by $k(\mathbf{x}, \mathbf{y}) = \langle S(\tilde{\mathbf{x}}), S(\tilde{\mathbf{y}}) \rangle$ and, similarly, the robust signature kernel $k_\Lambda(\mathbf{x}, \mathbf{y}) = \langle \Lambda(S(\tilde{\mathbf{x}})), \Lambda(S(\tilde{\mathbf{y}})) \rangle$ where $\Lambda$ is a tensor normalisation.

**Theorem C.1.** *Let $p \geq 1$, $\Lambda$ a tensor normalization, and $K \subset \mathcal{X}$ compact under $\alpha_p$. Then,*

    *(i) The signature kernel $k$ is universal to $\mathcal{F} = C(K, \mathbb{R})$ equipped with the uniform topology and characteristic to the dual $\mathcal{F}'$, the space of regular Borel measures on $K$.*

    *(ii) The robust signature kernel $k_\Lambda$ is universal to $\mathcal{F} = C_b(\mathcal{X}, \mathbb{R})$ equipped with the strict topology and characteristic to the dual $\mathcal{F}'$, the space of all finite Borel measures on $\mathcal{X}$.*

*Proof of Theorem C.1.* Part $(i)$ follows directly from the proof of Proposition 3.6 in [13]. For part $(ii)$ we shall proof that the feature map $F = \Lambda \circ S$ is universal and characteristic. The result then follows from Proposition 29 in [8]. We start by defining $\mathcal{P} = \mathcal{X}/\sim_t$ where the equivalence relation $\sim_t$ is defined in Appendix A.2. We equip $\mathcal{P}$ with the topology induced by the embedding $S : \mathcal{P} \to \tilde{T}((\mathbb{R}^{d+1}))$. By Proposition A.1, $F$ is a continuous and injective map from $\mathcal{P}$ into a bounded subset of $\tilde{T}((\mathbb{R}^{d+1}))$. Thus, $\mathcal{H} = \{\langle \ell, F \rangle \mid \ell \in T((\mathbb{R}^{d+1}))'\}$ is a subset of $\mathcal{F}$ that separates points. Furthermore, since $F$ takes values in the set of group-like elements, $\mathcal{H}$ is a subalgebra of $\mathcal{F}$ (under the shuffle product). It then follows from Theorem 7 and Theorem 9 in [8] that $F$ is universal and characteristic. The fact that $\mathcal{F}'$ is the space of all finite Borel measures on $\mathcal{X}$ is part (iii) of Theorem 9 in the same paper. Finally, as per Appendix A.3, the map $\mathbf{x} \mapsto \tilde{\mathbf{x}}$ is a continuous and injective embedding of $\mathcal{X}$ into $\mathcal{P}$ from which the result then follows. $\square$

With $d_k$ denoting the MMD for a given kernel $k : \mathcal{X} \times \mathcal{X} \to \mathbb{R}$, the following is a direct consequence of Theorem C.1.

**Corollary C.1.** *Let $p \geq 1$, $\Lambda$ a tensor normalization, and $K \subset \mathcal{X}$ compact under $\alpha_p$. Then, $d_k$ is a metric on $\mathcal{M}(K)$ and $d_{k_\Lambda}$ is a metric on $\mathcal{M}(\mathcal{X})$.*

---

$^5$If $\mathbf{x}$ is a càdlàg rough path, this is done via a *Young pairing* which results in a càdlàg $p$-rough path, $\tilde{\mathbf{x}}$, where the first level is given by $(x_t, t)$. For more information, we refer to Appendix A.3.

## D Forward sensitivities for SLIF network

In the general SSNN model, Theorem 3.2 gives the following result.

**Proposition D.1.** *Fix some weight $w_{ij} \in w$, a neuron $k \in [K]$ and let $\mathcal{G}_t^k$ denote the gradient of $(v^k, i^k)$ wrt. $w_{ij}$ at time $t$. Furthermore, define $\gamma : \{0, 1\} \to \mathbb{R}^2$ such that $\gamma_0 = (\mu_1, -\mu_2)w_{lk}$, $\gamma_1 = (\mu_1, 0)v_{reset}$, and let $\Gamma \in \mathbb{R}^{2 \times 2}$ be the drift matrix in the inter-spike SDE of $(v^k, i^k)$. Then,*

$$\mathcal{G}_t^k = e^{\Gamma(t-s)} \left( \mathcal{G}_s^k - \gamma_{\delta_{lk}} \partial_{w_{ij}} s + \delta_{il}\delta_{jk} e_2 \right), \tag{19}$$

*where $e_n \in \mathbb{R}_2$ is the n'th unit vector, $l$ is the neuron in $\mathrm{Pa}_k \cup \{k\}$ with the most recent spike time before $t$, and we denote this spike time by $s$. If $t$ is a spike time of neuron $k$ it therefore follows that*

$$\partial_{w_{ij}} t = \frac{\lambda(v_{t_{prev}}^k)\partial_{w_{ij}} t_{prev} - \int_{t_{prev}}^t \nabla\lambda(v_r^k)e_1^T\mathcal{G}_r^k dr}{\lambda(v_t^k)}, \tag{20}$$

*where $t_{prev}$ is the previous spike time of neuron $k$. In the case of a deterministic SNN, formula (20) is replaced by*

$$\partial_{w_{ij}} t = -\frac{e_1^T\mathcal{G}_t^k}{\mu_1(i_t^k - v_t^k)}. \tag{21}$$

*Proof.* Throughout we fix some $t > 0$ and let $s < t$ denote most recent event time preceding $t$ with $l$ the index of the neuron firing at time $s$. We define the process $dw_t = 0dt$ with $w_0 = w_{ij}$ and with a slight abuse of notation we shall write $y_t^k = (v_t^k, i_t^k, s_t^k, w_t)$. We will leave out the event index $n$ for notational simplicity. Since $y_t^k$ depends on $y_s$ only through $y_s^k$ and $\nabla\mathcal{T}_l$ is block diagonal, a direct consequence of eq. (7) is

$$\mathcal{G}_t^k = (I \quad 0) \, \partial_{y_s^k} y_t^k \left( \nabla\mathcal{T}_j^l(y_{s-}^k)\partial_{w_{ij}} y_{s-}^k - \left( \mu(y_s^k) - \nabla\mathcal{T}_l^k(y_{s-}^k)\mu(y_{s-}^k) \right) \partial_{w_{ij}} s \right)$$

where $\mu(v, i, s, w) = (\mu_1(i - v), -\mu_2 i, \lambda(v), 0)$. If $l \in \mathrm{Pa}_k \cup \{k\}$, then $\mathcal{T}_l^k = $ id and therefore $\mathcal{G}_t^k = (I \quad 0) \, \partial_{y_s^k} y_t^k \partial_{w_{ij}} y_s^k$. One can then reapply the formula above until $l \in \mathrm{Pa}_k \cup \{k\}$. By the flow property, it follows that we may assume without loss of generality that $l \in \mathrm{Pa}_k \cup \{k\}$. This leaves us with two cases. We define $z_t^k = (v_t^k, i_t^k)$ so that $(I \quad 0) \, \partial_{y_s^k} y_t^k = \partial_{z_s^k} z_t^k$ and $\partial_{w_{ij}} z_t^k = \mathcal{G}_t^k$. Furthermore, let $a = \delta_{il}\delta_{jk}$.

*Case 1, $l \in \mathrm{Pa}_k$:* In this case $\mathcal{T}_l^k(v, i, s, w) = (v, i + aw + (1 - a)c, s, w)$ where $c$ is a constant. As a result

$$\partial_{z_s^k} z_t^k \nabla\mathcal{T}_l^k(y_{s-}^k)\partial y_{s-}^k = \partial_{z_s^k} z_t^k \mathcal{G}_t^k + a\partial_{i_s^k} z_t^k,$$
$$\partial_{z_s^k} z_t^k \left( \mu(y_s^k) - \nabla\mathcal{T}_l^k(y_{s-}^k)\mu(y_{s-}^k) \right) = \partial_{z_s^k} z_t^k \gamma_0,$$

In total,

$$\mathcal{G}_t^k = \partial_{z_s^k} z_t^k \left( \mathcal{G}_t^k - \gamma_0\partial_{w_{ij}} s + ae_2 \right).$$

*Case 2, $l = k$:* In this case $\mathcal{T}_l^k(v, i, s, w) = (v - v_{reset}, i, \log u - \alpha, w)$ so that

$$\partial_{z_s^k} z_t^k \nabla\mathcal{T}_l^k(y_{s-}^k)\partial y_{s-}^k = \partial_{z_s^k} z_t^k \mathcal{G}_t^k,$$
$$\partial_{z_s^k} z_t^k \left( \mu(y_s^k) - \nabla\mathcal{T}_l^k(y_{s-}^k)\mu(y_{s-}^k) \right) = \partial_{z_s^k} z_t^k \gamma_1,$$

and, thus,

$$\mathcal{G}_t^k = \partial_{z_s^k} z_t^k \mathcal{G}_t^k - \partial_{z_s^k} z_t^k \gamma_0\partial_{w_{ij}} s.$$

Note that $z_t^k$ is an Ornstein-Uhlenbeck process initialized at $z_s^k$ and with drift and diffusion matrices

$$\Gamma = \begin{pmatrix} -\mu_1 & \mu_1 \\ 0 & -\mu_2 \end{pmatrix}, \quad \Sigma = \begin{pmatrix} \sigma_1 & 0 \\ 0 & \sigma_2 \end{pmatrix}.$$

As a result, we can directly compute $\partial_{z_s^k} z_t^k = e^{(t-s)\Gamma}$. This proves that eq. (19) holds. Eq. (20) then follows directly from (6) and the fact that $\mathcal{E}_k(y) = s^k$. $\qquad\square$

From this the results of Section 4.4 follow since the terms $\partial_{w_{ij}} s$ vanish whenever $s$ is the spike time of a neuron $l$ that is not a descendant of neuron $j$. Thus, equation (19) only includes terms depending on the activity of the pre- and post-synaptic neuron. In particular, there is no need to store the gradient path $\mathcal{G}_t^k$ for each combination of neuron $k$ and synapse $ij$, but each neuron only needs to keep track of the paths for its incoming synapses. This reduces the memory requirements from the order of $K^3$ to only $K^2$ (which is needed anyway to store the weight matrix). In general, the gradient paths can be approximated by simply omitting the terms $\partial_{w_{ij}} s$.

# E  Experiments

## E.1  Input current estimation

[6]For each combination of sample size and $\sigma$ we sample a data set of spike trains using Algorithm 1 with $N = 3$, i.e., up until the first three spikes are generated. We use `diffrax` to solve the inter-Event SDE with a step size of 0.01 and the numerical solver is the simple Euler-Maruyama method. We then sample an initial guess $c \sim \text{Unif}([0.5, 2.5])$ and run stochastic gradient descent using the approach described in 4.1. That is, for each step, we generate a batch of the same size as the sample size and use $d_k$ to compare the generated batch to the data. For each step we also compare the absolute error between the average spike time of the first three spikes of the generated sample to a hold a out test set of the same size as the sample. We use the RMSProp algorithm with a decay rate of 0.7 and a momentum of 0.3 which we found to work well in practice. The learning rate is 0.001. The experiment was run locally on CPU with an Apple M1 Pro chip with 8 cores and 32 GB of ram. The entire experiment took approximately 3-6 hours to run. For the exact details of this experiment we refer to the notebook `snnax/notebooks/single_neuron.ipynb` in the supplementary material.

## E.2  Synaptic weight estimation

As above, for each sample size $D \in \{256, 512, 1024\}$ we sample a data set of spike trains using Algorithm 1 with $T = 1$ and with the same differential equation solver setup as above. Thus, in this case, the number of spikes varies across each sample path. The parameters are chosen as follows:

- $v_{reset} = 1.2$
- $\lambda(v) = \exp(5(v - 1)$
- $\mu = (6, 5)$
- $\sigma = I_2/4$

For each sample size the data was generated using the same randomly sampled weight matrix $w$ which represents a feed-forward network of the dimensions described in Section 4 and which was constructed as follows: for the weight matrix from layer $l$ to layer $l + 1$, say $w^l$, we sample each entry from $\text{Unif}([0.5, 1.5])$ and then normalize by $3/K_l$ where $K_l$ is the number of neurons in layer $l$. The normalisation makes sure that the spike rate for the neurons in each layer is appropriate.

For each data set (each sample size) we then train a spiking neural net of the same network structure to match the observed spike trains. This is done using stochastic gradient descent with a batch size of $B = 128$ and by computing $d_k$ on a generated batch and a batch sampled from the data set at each step. In order to avoid local minimums[7] we match the number of spikes between the generated spike trains and the ones sampled from the data set. Also, we sample from the data set without replacement so that we loop through the whole data set every $D/B$ steps. We run RMSProp for 1500 steps with a momentum of 0.3 and a learning rate of 0.003 for the first 1000 steps and 0.001 for the last 500 steps.

This experiment was run in the cloud using Azure AI Machine Learning Studio on a NVIDIA Tesla V100 GPU with 6 cores and 112 GB of RAM. The entire experiment took around 12-16 hours to run. For the exact details we refer to the notebook `snnax/notebooks/spiking_neural_net.ipynb` in the supplementary material.

---

[6]For an updated version of some of the experiments and a general implementation of SSNNs in `diffrax` we refer to github.com/cholberg/snnax.

[7]Note that the loss landscape is inherently discontinuous since whenever the parameters are altered in such a way that an additional spike appears, the expected signature will jump.

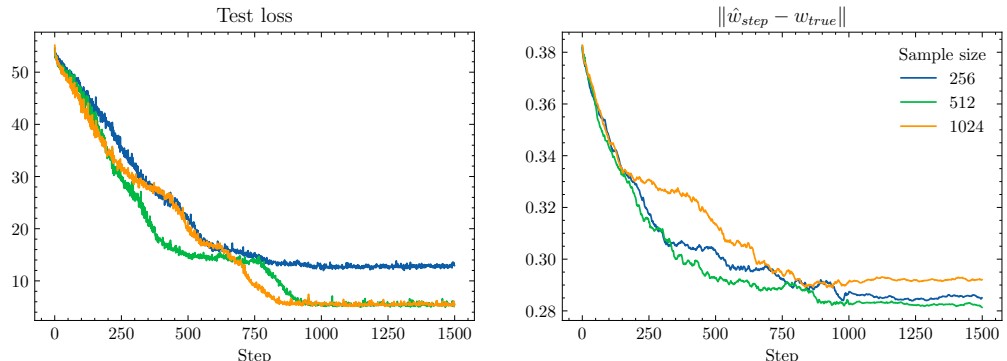

Figure 2: We estimate the synaptic weights $w$ across three different sample sizes using the signature kernel MMD truncated at depth 3 and stochastic gradient descent with a batch size of 128. On the left we report the loss on a hold out test set. On the right is the mean absolute error between the entries of the currently estimated weight matrix $\hat{w}_{step}$ and the true weight matrix $w_{true}$.

