# OpenReview forum: "Exact Gradients for Stochastic Spiking Neural Networks Driven by Rough Signals"
_NeurIPS.cc/2024/Conference — NeurIPS 2024 poster_

### Official Review · Reviewer_2eDR · 2024-07-06

**Soundness:** 3
**Presentation:** 3
**Contribution:** 3
**Rating:** 7
**Confidence:** 4

**Summary:**

Spiking neural networks face the problem of non-differentiability of loss function due to the Heaviside activation function, which works as the spiking function. To back-propagate the loss through the network, Heaviside is replaced with surrogate functions as a workaround.

The paper establishes the theoretical existence of the exact gradient for stochastic spiking neural networks, where stochasticity is introduced in spike timing, and the gradient of spike timing is computed with respect to the model parameters. It uses the existing theories of stochastic differential equations (rough path theory) to handle discontinuous spiking events and introduces a novel kernel (Marcus signature kernel) to handle such discontinuities. They further provide an algorithm that can perform parameter estimation of a two-layer network on toy examples.

**Strengths:**

The paper uses stochastic leaky-intergrate-and-fire neurons, which have randomness in the evolution of membrane potential as well as spike timings. It uses the mathematical framework of rough path theory to obtain the gradient of the spike timings with respect to network parameters and formally prove the existence of such a gradient in this framework. The paper is well-written and provides intuitive descriptions along with rigorous mathematical proofs. It brings a novel theoretical insight into spiking neural network training.

**Weaknesses:**

The paper demonstrates it implementation through a toy parameter estimation problem on an one-hidden layer feed forward network of dimension $4 \times 16 \times 2$, where the network parameters are estimated. Such a network, having a tiny input dimension, is not readily applicable to realistic datasets. Further, rather than implementing the algorithm as a parameter search (as done in standard SNN training), it is implemented as a parameter estimation problem. The algorithmic aspects of the paper are somewhat open; the paper discusses several possible avenues, such as "discretize-then-optimize" vs. "optimize-then-discretize" frameworks. The authors must discuss the computation complexity of the proposed EventSDESolve algorithm, which uses the discretize-then-optimize approach.

**Questions:**

**Doubt:** Assumptions 3.3, 3.4 and 3.5 require the event and transition function to be continuously differentiable [line 194]. How do such functions model discontinuous spikes? In contrast, line 229 mentions that trajectories and spike times are almost surely differentiable. Please clarify.

**Measure of Randomness:** It is shown that the zeroth-order derivative of the Heaviside function approximates a surrogate function depending upon the noise distribution [1]. How, in the present case, does the variance of the introduced noise (in the Brownian motion) affect the exact gradient? Does it recover the gradient of Heaviside in a limiting case? Which of the two noises (Brownian motion vs. transition noise [line 166]) used in the SLIF modelling is essential to obtain the exact gradient -- are either of them a mathematical convenience?

**Computational Feasibility:** Please discuss the computational complexity of EventSDESolve. How does the computational complexity of the gradient scale with the number of parameters? Can the algorithm handle real datasets? Why did the authors choose to solve a parameter estimation problem (through estimation of input current) instead of standard training of the network parameters?

[1] Mukhoty, Bhaskar, et al. "Direct training of snn using local zeroth order method." Advances in Neural Information Processing Systems 36 (2024).

**Limitations:**

Yes, authors do discuss some of the limitations of their work.

---

> ### Author Rebuttal · Authors · 2024-08-07
>
> We thank the reviewer for taking the time to read the paper and provide constructive feedback. We hope that the points below in combination with the author rebuttal answers all the questions that the reviewer may still have.
>
> **W**: The paper demonstrates it implementation through a toy parameter estimation problem on an one-hidden layer feed forward network of dimension $4\times 16 \times 2$, where the network parameters are estimated. Such a network, having a tiny input dimension, is not readily applicable to realistic datasets. Further, rather than implementing the algorithm as a parameter search (as done in standard SNN training), it is implemented as a parameter estimation problem. The algorithmic aspects of the paper are somewhat open; the paper discusses several possible avenues, such as "discretize-then-optimize" vs. "optimize-then-discretize" frameworks. The authors must discuss the computation complexity of the proposed EventSDESolve algorithm, which uses the discretize-then-optimize approach.
>
> **A**: We thank the reviewer for the suggestion and agree that this would be a good addition to the paper. See also point 2 of the author rebuttal.
>
>
> **Q**: **Doubt**: Assumptions 3.3, 3.4 and 3.5 require the event and transition function to be continuously differentiable [line 194]. How do such functions model discontinuous spikes? In contrast, line 229 mentions that trajectories and spike times are almost surely differentiable. Please clarify.
>
> **A**: The event and transition functions determine when the discontinuity happens and how the system jumps at the discontinuity respectively. These functions need to be continuous (as functions of the state) in order for the event times and the solution trajectories to be differentiable (wrt. the initial condition). Note that the discontinuity we are modelling is a discontinuity wrt. time, i.e., the state jumps at certain points in time when an event is triggered. The exact timing and size of these jumps, however, is still allowed to be smooth (as a function of the parameters/initial condition).
>
>
> **Q**: **Measure of Randomness**: It is shown that the zeroth-order derivative of the Heaviside function approximates a surrogate function depending upon the noise distribution [1]. How, in the present case, does the variance of the introduced noise (in the Brownian motion) affect the exact gradient? Does it recover the gradient of Heaviside in a limiting case? Which of the two noises (Brownian motion vs. transition noise [line 166]) used in the SLIF modelling is essential to obtain the exact gradient -- are either of them a mathematical convenience?
>
> **A**: The noise from the Brownian motion is implicitly part of eqs. (6) and (7) in two ways: 1) Both expressions depend on the solution trajectories $y^{n-1}_t$ which, in turn, are noisy due to the presence of the Brownian motion. 2) $\partial y^n_t$ can be shown to satisfy another SDE which in general has a diffusion term as well driven by the same Brownian motion.
>
> In the deterministic limit we recover the exact gradients of the deterministic SNN with firing exactly at the threshold.
>
> In the absence of a diffusion term, the dynamics reduce to an ODE in which case exact gradients always exist. In the context of our paper, this is because Assumption 3.4 is satisfied in this case. On the other hand, if a diffusion term is present, we require stochastic firing in order to ensure existence of derivatives via Assumption 3.4.
>
>
> **Q**: **Computational Feasibility**: Please discuss the computational complexity of EventSDESolve. How does the computational complexity of the gradient scale with the number of parameters? Can the algorithm handle real datasets? Why did the authors choose to solve a parameter estimation problem (through estimation of input current) instead of standard training of the network parameters?
>
> **A**: We believe that this question is answered by points 2 and 3 of the author rebuttal.

---

> > ### Comment · Reviewer_2eDR · 2024-08-12
> >
> > I thank the authors for the clarification presented. It could be beneficial for future readers if some of the concerns raised by the reviewers and the corresponding responses could be summarized in the paper's appendix.

---

### Official Review · Reviewer_ywki · 2024-07-09

**Soundness:** 4
**Presentation:** 2
**Contribution:** 3
**Rating:** 6
**Confidence:** 2

**Summary:**

This work develops a mathematical framwork to compute the gradients of stochastic spiking neural networks, or more generally Event SDEs. The proposed framework is an alternative to existing surrogate gradient frameworks and an extension of prior adjoint based work (e.g. EventProp) in the presence of stochasticity. Jax-based code is provided with the submission.

**Strengths:**

The manuscript offers an attractive solution to the problem of solving SNNs, based on rigorous demonstrations. The derivations and description makes too much use of very specific jargon, making it difficult to evaluate with respect to the existing methods. At this point, if there is indeed practical merit in the proposed approach, it is unlikely to have an impact to the field as is. The lack of testing on standard SNN benchmarks doesn't help this problem. I believe these problems could be fixed by making more parallels with existing work and at least ont benchmark that demonstrates the effectiveness of the proposed method. The yin yang dataset in [55] would be a good candidate.

- Mathematically rigorous demonstration of stochastic SNN gradients is novel and relevant. Strickly speaking, existing methods already demonstrated exact gradients in SNNs in the absence of gradients and when events are known [55], but the extension to stochastic inputs and dynamics is new to my knowledge. Note that in this sense, the statement L260 is already no longer popular belief.
- Connection to three-factor rules in the feed-forward case extends prior work in the field to stochastic models

**Weaknesses:**

- Use of jargon and mathematical assumptions which are sometimes not sufficiently connected with the studied problem. This is a missed opportunity IMO to connect with the relevant communities. e.g. I have little clue what this means:
  "train NSDEs non-adversarially using a class of maximum mean discrepancies (MMD) endowed with signature kernels [51] indexed on spaces of continuous paths as discriminators".
 Why adversarial? what is a continuous signature kernel?
 Maybe I'm the wrong reader for this paper, but then this would likely apply to most readers interested in training SNNs.

- Lack of a benchmark and experimental comparison to prior work based on eventprop or surrogate gradients. The framework is written in jax, and should in principle allow a scalable simulation, why not run on a simple SNN benchmark? The provided experiments are overly simple and leaving this open as a vague limitation statement is not enough.

**Questions:**

- Please explain some ambiguous sentences and less known facts:
  - L211: "If the diffusion term is absent we recover the gradients in [6]" from which expression to we recover the gradients in [6]?
  - Give at least a one sentence description of cadlag (right continuous, left limit). Why is it relevant to the modeling of SNNs?
  - What is a continuous signature kernel? Can you give an intuition without having to read another technical paper?
- Definition 3.1and its eq (3-5) are to my understanding a general version of eq (2). Is it really necessary to define these two models separately? I would recommend using the same notation if possible, or to the very least to connect the expressions across the two definitions.
- Can you solve at least one practical SNN training problem?
- How does the training effort (wall time, memory) compare to existing methods? Is this an impediment to test on a benchmark?

**Limitations:**

Some limitations are discussed in the conclusions, but leaving a simple SNN benchmark to future work is not a limitation, but a shortcoming of the manuscript in my opinion.

---

> ### Author Rebuttal · Authors · 2024-08-07
>
> We thank the reviewer for taking the time to read the paper and provide constructive feedback. We hope that the points below in combination with the author rebuttal answers all the questions that the reviewer may still have.
>
> **W**: Use of jargon and mathematical assumptions which are sometimes not sufficiently connected with the studied problem. This is a missed opportunity IMO to connect with the relevant communities. e.g. I have little clue what this means: "train NSDEs non-adversarially using a class of maximum mean discrepancies (MMD) endowed with signature kernels [51] indexed on spaces of continuous paths as discriminators". Why adversarial? what is a continuous signature kernel? Maybe I'm the wrong reader for this paper, but then this would likely apply to most readers interested in training SNNs.
>
> **A**: We tried our best to merge two very distant communities: Those of computational neuroscience and rough analysis. We believe the value of the contribution is also in this attempt. Due to space limits we cannot go into further details, but we can add the following clarification to the camera-ready version: Training a Neural Stochastic Differential Equation (NSDE) involves minimizing the distance between the path-space distribution generated by the SDE and the empirical distribution supported by observed data sample paths. Various training mechanisms have been proposed in the literature. State-of-the-art performance has been achieved by training NSDEs adversarially as Wasserstein-GANs. However, GAN training is notoriously unstable, often suffering from mode collapse and requiring specialized techniques such as weight clipping and gradient penalty. In [51], the authors introduce a novel class of scoring rules based on signature kernels, a type of characteristic kernel on paths, and use them as the objective for training NSDEs non-adversarially.
>
>
> **W**: Lack of a benchmark and experimental comparison to prior work based on eventprop or surrogate gradients. The framework is written in jax, and should in principle allow a scalable simulation, why not run on a simple SNN benchmark? The provided experiments are overly simple and leaving this open as a vague limitation statement is not enough.
>
> **A**: Please see our points 2 and 3 of the author rebuttal above.
>
>
> **Q**: Please explain some ambiguous sentences and less known facts:
>
> - L211: "If the diffusion term is absent we recover the gradients in [6]" from which expression to we recover the gradients in [6]?
> **A**: By this we mean that their eq. (9) is equivalent to our eq. (6). We suggest to add the following in Remark 3.1: "In particular, eq. (6) for \(n=1\) is exactly eq. (9) in [6]." Note, however, that they only consider the derivative of the first event time and they explicitly model the dependence on \(t\) in the event function. We chose to assume that the event function only depends on the state \(y\). (This is only for notational simplicity since \(t\) can easily be included in the state if need be.)
>
> - Give at least a one sentence description of cadlag (right continuous, left limit). Why is it relevant to the modeling of SNNs?
> **A**: We suggest to add above description as a footnote to the first mention of càdlàg in the main body of text. We note that trajectories of (S)SNNs exhibit jumps due to discontinuous event-triggering and therefore are naturally modelled as càdlàg paths. Spike trains can be viewed as càdlàg step functions counting the number of spikes over time, the membrane potential is right continuous and has left limits (at the discontinuity points the left limit is equal to the firing threshold).
>
> - What is a continuous signature kernel? Can you give an intuition without having to read another technical paper?
> **A**: The signature transform can be viewed as feature map on path space defined as an infinite series of iterated integrals. These iterated integrals play the same role as tensor monomials for vectors on $\mathbb R^d$, so the signature can be thought as the analogue of a Taylor expansion on path space. The (continuous) signature kernel of two paths is defined as an inner product of their signatures. It serves as a natural measure of (dis)similarity between two curves [*].  We propose to add this explanation to Section 4.1. in the camera-ready version of the paper.
>
> [*] Király, Franz J., and Harald Oberhauser. "Kernels for sequentially ordered data." Journal of Machine Learning Research 20.31 (2019): 1-45.
>
>
> **Q**: Definition 3.1 and its eq (3-5) are to my understanding a general version of eq (2). Is it really necessary to define these two models separately? I would recommend using the same notation if possible, or to the very least to connect the expressions across the two definitions.
>
> **A**: We understand the confusion that this might cause and would like to point to the paragraph beginning on L219 where we explain the connection between SSNNs and Def. 3.1. The reason for choosing this more general framework is because all of the theory that we develop holds in full generality to what we call Event RDEs (as defined in Section A.4 of the Appendix). This includes many more use cases than SNNs and, in particular, is a generalization (and formalization) of the settings discussed in e.g. [6, 25].
>
>
> **Q**: Can you solve at least one practical SNN training problem?
>
> **A**: Please see the response and suggestions in points 2 and 3 of the author rebuttal.
>
>
> **Q**: How does the training effort (wall time, memory) compare to existing methods? Is this an impediment to test on a benchmark?
>
> **A**: We refer to points 2 and 3 of the author rebuttal.

---

> > ### Comment · Reviewer_ywki · 2024-08-12
> >
> > Thanks you for the clarifications, I maintain my score.

---

### Official Review · Reviewer_sP65 · 2024-07-10

**Soundness:** 3
**Presentation:** 3
**Contribution:** 3
**Rating:** 6
**Confidence:** 2

**Summary:**

The paper introduces a mathematical framework using rough path theory to model stochastic spiking neural networks as stochastic differential equations with event discontinuities, driven by càdlàg rough paths.  This framework accommodates potential jumps in both solution trajectories and driving noise. Furthermore, the authors introduce Marcus signature kernels to extend continuous signature kernels to càdlàg rough paths and using them to define a general-purpose loss function for training stochastic SNNs as generative models.

**Strengths:**

1. The paper is generally well-written. Despite its technical nature and numerous technical definitions, it is clear that the authors have made efforts to present the material in an accessible way.

2. The framework in the paper identifies sufficient conditions for the existence of pathwise gradients of solution trajectories and event times with respect to network parameters, and deriving a recursive relation for these gradients.

**Weaknesses:**

I am not an expert in rough path theory and I don't see any major weaknesses.

**Questions:**

1. How does the performance of the proposed gradient-based training compare to other established methods in the literature, such as those in [1,2,3]? More exhaustive empirical evaluations would be valuable and this would help to understand the benefits and limitations of this approach in practice.

2. Can the framework be extended to the case of single spike scenario (for instance in [1,2,3])?

[1] Göltz, J., Kriener, L., Baumbach, A. et al. Fast and energy-efficient neuromorphic deep learning with first-spike times. Nat Mach Intell 3, 823–835 (2021).

[2] I. M. Comsa, K. Potempa, L. Versari, T. Fischbacher, A. Gesmundo and J. Alakuijala, "Temporal Coding in Spiking Neural Networks with Alpha Synaptic Function," ICASSP 2020 - 2020 IEEE International Conference on Acoustics, Speech and Signal Processing (ICASSP), Barcelona, Spain, 2020, pp. 8529-8533, doi: 10.1109/ICASSP40776.2020.9053856.

[3] Stanojević, Ana et al. “An Exact Mapping From ReLU Networks to Spiking Neural Networks.” Neural networks : the official journal of the International Neural Network Society 168 (2022): 74-88.

**Limitations:**

There is no potential negative societal impact.

---

> ### Author Rebuttal · Authors · 2024-08-07
>
> We thank the reviewer for taking the time to read the paper and provide constructive feedback. We hope that the points below in combination with the author rebuttal answers all the questions that the reviewer may still have.
>
> **Q**: How does the performance of the proposed gradient-based training compare to other established methods in the literature, such as those in [1,2,3]? More exhaustive empirical evaluations would be valuable and this would help to understand the benefits and limitations of this approach in practice.
>
> **A**: We note that benchmarking is limited by the fact that, to our knowledge, we are the first to consider stochastic SNNs (in the sense that the ODEs are replaced with SDEs). We agree that more exhaustive empirical evaluations would be very valuable and refer to point 2 and 3 of the author rebuttal above.
>
>
> **Q**: Can the framework be extended to the case of single spike scenario (for instance in [1,2,3])?
>
> **A**: Yes, there is nothing prohibiting us from applying the framework to single spike scenarios. In particular: 1) The main theorem simply states the existence of gradients and how one would compute them when having access to exact solutions of the inter-event SDE. Thus, this result also holds for the first spike times of an output layer of neurons. 2) The signature kernel works for any collection of càdlàg paths. In principle one could just take the first spike time of each output neuron and convert these to spike trains to be fed into the signature kernel MMD.

---

> > ### Comment · Reviewer_sP65 · 2024-08-13
> >
> > Thank you for your comments and rebuttal. It would be valuable to include the points in the general response from the authors in the paper for additional context. I will keep my score.

---

### Official Review · Reviewer_NV97 · 2024-07-11

**Soundness:** 3
**Presentation:** 3
**Contribution:** 3
**Rating:** 6
**Confidence:** 2

**Summary:**

This paper introduces a general mathematical framework to model stochastic spiking neural networks (SSNN) as stochastic differential equations with event discontinuities, and identifies sufficient conditions ensuring the existence of gradients. With a newly defined loss function, SSNNs can be trained as generative models with an end-to-end autodifferentiable solver. Some empirical verifications demonstrate the effectiveness of the proposed method, and also, discussions on the connection with bio-plausible learning algorithms are provided.

**Strengths:**

1. This paper proposes a mathematically rigorous framework to model SSNNs as SDEs with event discontinuities and driven by cadlag rough paths. The paper also identifies sufficient conditions for the existence of gradients, which strictly generalizes previous results only considering ODEs.

2. This paper provides the first gradient-based training of a large class of SSNNs with noise processes in both the spike timing and the network’s dynamics.

3. This paper discusses how the results lead to bio-plausible learning algorithms.

**Weaknesses:**

1. There are several assumptions in the analysis. It would be better to have more discussions on whether these assumptions (e.g., Assumptions 3.1 and 3.2) can hold in common conditions.

2. This paper only conducts experiments on two toy settings. It would be better to discuss how to apply the method to more application conditions.

**Questions:**

What is the computational complexity of SDESolveStep and RootFind?

**Limitations:**

The authors discussed limitations in Section 5.

---

> ### Author Rebuttal · Authors · 2024-08-07
>
> We thank the reviewer for taking the time to read the paper and provide constructive feedback. We hope that the points below in combination with the author rebuttal answers all the questions that the reviewer may still have.
>
> **W**: There are several assumptions in the analysis. It would be better to have more discussions on whether these assumptions (e.g., Assumptions 3.1 and 3.2) can hold in common conditions.
>
> **A**: We agree that we could have spent a little more effort in justifying these assumptions and had also done so in an earlier version, but chose not to here due to space constraints. We propose to make the changes suggested in point 1 of the author rebuttal.
>
>
> **W**: This paper only conducts experiments on two toy settings. It would be better to discuss how to apply the method to more application conditions.
>
> **A**: We agree that the two settings considered in the paper are very limited. The main focus of the paper lies in establishing Theorem 3.1 and explaining its relevance in the context of SSNNs. The algorithm that we present based on the discretize-then-optimize approach should be viewed as a first step in applying these results. Finding ways in which to modify this algorithm to allow for better scaling is left open as an avenue to explore in future work. See also points 2 and 3 of the author rebuttal above for more details and with our suggestions.
>
> **Q**: What is the computational complexity of SDESolveStep and RootFind?
>
> **A**: It is hard to state in generality what the computational complexity of the two operations are since they both depend on the vector field and the method employed. For example, a simple Euler step of the solver requires only a single evaluation of the vector field whereas higher order solves would require more complex operations. We can, however, say something more specific on how the EventSDESolve algorithm scales in the case of a SSNN. See points 2 and 3 of our general rebuttal and the suggestion given therein. The main takeaway is that scaling the discretize-then-optimize algorithm up to a high number of neurons is generally difficult. For these tasks it would appear that the optimize-then-discretize approach is more suitable. Due to time constraints, we have not been able to test out this approach, but as mentioned in the paper, the adjoint equation for the gradients follow readily from Theorem 3.1. We can add a section in the appendix detailing the optimize-then-discretize approach as suggested in point 3 of the author rebuttal.

---

> > ### Comment · Reviewer_NV97 · 2024-08-12
> >
> > I would like to thank the authors for their detailed responses and discussions, and I keep my score.

---

### Author Rebuttal · Authors · 2024-08-07

We would like to thank all the reviewers for taking the time to go through the paper and providing valuable feedback. We agree with many of the points that have been raised and believe that most, if not all, can be accommodated in a camera-ready version. Apart from minor points and clarifications, we found the main three weaknesses to be shared among all reviews:

1. **Assumptions**: We could have spent more time justifying the assumptions needed to prove existence of solutions of Event SDEs as well as for the main theorem. While we briefly discuss all assumptions in the context of SSNNs, we agree that more could be done. In particular, we agree that the first two Assumptions deserve more explanation and would be happy to add this in the final version. We suggest to add the following after L173: "Assumptions 3.1 and 3.2 simply ensure that an event cannot be triggered immediately upon transitioning. This holds in many settings of interest. For example, for the usual deterministic LIF neuron, $\textnormal{im } \mathcal{T} = 0$ and $\ker \mathcal{E} = 1$ and the length of the refractory period is directly linked to $c$ in Assumption 3.1".  We note that these assumptions are standard in the literature concerning stochastic hybrid processes (see, e.g., our references [31, 32]). We also note that all assumptions are discussed in the context of (S)SNNs after Remark 3.2.

2. **Computational complexity**: More clarity is needed regarding the complexity of the provided algorithm and its alternatives. This is a fair point and we believe it can be answered by adding an additional remark after Remark 3.4 stating the following: "One the one hand, the EventSDESolve algorithm as presented here scales poorly in the number of events since it requires doing a full SDESolve and an additional RootFind each time an event occurs. This problem becomes especially prevalent for SSNNs with a large number of neurons since in this case an event is triggered every time a single neuron spikes and the inter-spike SDE that needs to be solved is high-dimensional. On the other hand, there are multiple ways to mitigate this issue. Firstly, one could relax the root-finding step and simply trigger a spike as soon as $e\ge 0$ and take this as the spike time. For the backward pass one could then solve the adjoint equations (for which you need need to store the spike times in the forward pass). The resulting algorithm would be similar to the one derived in [55] for deterministic SNNs. Secondly, for special architectures such as a feed-forward network, given the spikes from the previous layer, one could solve the EventSDE for each neuron in the current layer independently of all other neurons. This would imply that a forward (or backward) pass of the entire SSNN scales as $O(KS)$ where $S$ is the cost of the forward (or backward) pass of a single neuron and $K$ is the number of neurons."

3. **Examples**: The primary objective of the paper is to lay out the theoretical foundations of gradient-based learning with stochastic SNNs. Although we provided an initial implementation, which is well-suited for low dimensional examples, a robust version that scales to a high number of neurons is beyond the scope of the paper. Examples that require a much higher number of neurons than the two examples already discussed will be hard to handle with the discretize-then-optimize approach for the reasons given above. We propose to add an additional section in the Appendix where we give a derivation for the adjoint equations in a SSNN and provide pseudo-code for an optimize-then-discretize algorithm. As a final comment, we note that the provided Algorithm works for any given event SDE.

---

### Decision · Program_Chairs · 2024-09-25

**Decision:**

Accept (poster)

**Comment:**

This paper describes a novel formalism for computing gradients within spiking neural networks using stochastic differential equations.  All four reviewers felt that it was above the threshold for acceptance, and I'm pleased to report that it has been accepted to NeurIPS.  Congratulations!  Please revise the manuscript according to the reviewer comments and discussion points